# Parameter Averaging for SGD Stabilizes the Implicit Bias towards Flat Regions

## Abstract

Stochastic gradient descent is a workhorse for training deep neural networks due to its excellent generalization performance. Several studies demonstrated this success is attributed to the implicit bias of the method that prefers a flat minimum and developed new methods based on this perspective. Recently, Izmailov et al. (2018) empirically observed that an averaged stochastic gradient descent with a large step size can bring out the implicit bias more effectively and can converge more stably to a flat minimum than the vanilla stochastic gradient descent. In our work, we theoretically justify this observation by showing that the averaging scheme improves the *bias-optimization tradeoff* coming from the stochastic gradient noise: a large step size amplifies the bias but makes convergence unstable, and vice versa. Specifically, we show that the averaged stochastic gradient descent can get closer to a solution of a penalized objective on the sharpness than the vanilla stochastic gradient descent using the same step size under certain conditions. In experiments, we verify our theory and show this learning scheme significantly improves performance.

## 1 Introduction

Stochastic gradient descent (SGD) (Robbins & Monro, 1951) is a powerful learning method for training modern deep neural networks. In order to further improve the performance, a great deal of SGD variants such as adaptive gradient methods has been developed. However, SGD is still the workhorse because SGD often generalizes better than these variants even when they achieve much faster convergence regarding the training loss (Keskar & Socher, 2017; Wilson et al., 2017; Luo et al., 2019). Therefore, the study of the *implicit bias* of SGD, explaining why it works so better, is nowadays an active research subject.

Among such studies, *flat minima* (Hochreiter & Schmidhuber, 1997) has been recognized as an important notion relevant to the generalization performance of deep neural networks, and SGD has been considered to have a bias towards a flat minimum. Hochreiter & Schmidhuber (1997); Keskar et al. (2017) suggested the correlation between flatness (sharpness) and generalization, that is, flat minima generalizes well compared to sharp minima, and Neyshabur et al. (2017) rigorously supported this correlation under $\ell_2$-regularization by using the PAC-Bayesian framework (McAllester, 1998; 1999). Furthermore, by the large scale experiments, Jiang et al. (2020) verified that the flatness measures reliably capture the generalization performance and are the most relevant among 40 complexity measures. In parallel, Keskar et al. (2017) empirically demonstrated that SGD prefers a flat minimum due to its own stochastic gradient noise and subsequent studies (Kleinberg et al., 2018; Zhou et al., 2020) proved this implicit bias based on the smoothing effect due to the noise and stochastic differential equation, respectively.

Along this line of research, there are endeavors to enhance the bias aiming to improve performance. Especially, stochastic weight averaging (SWA) (Izmailov et al., 2018) and sharpness aware minimization (SAM) (Foret et al., 2020) achieved significant improvement in generalization performance over SGD. SWA is a cyclic averaging scheme for SGD, which includes the averaged SGD (Ruppert, 1988; Polyak & Juditsky, 1992) as a special case. Averaged SGD with an appropriately small step size or diminishing step size to zero is well known to be an efficient method that achieves statistically optimal convergence rates for the convex optimization problems (Bach & Moulines, 2011; Lacoste-Julien et al., 2012; Rakhlin et al., 2012). However, such a small step size strategy does not seem useful for

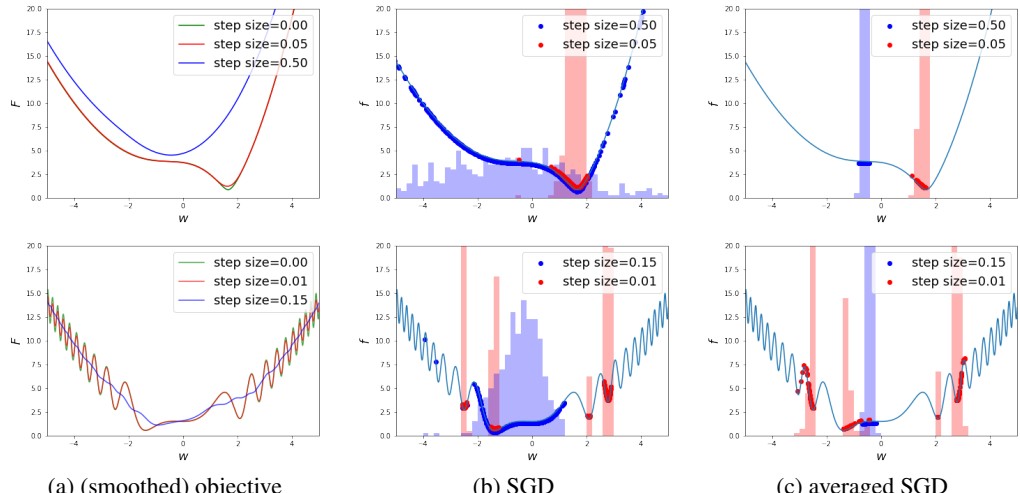

(a) (smoothed) objective      (b) SGD      (c) averaged SGD

Figure 1: We run SGD and averaged SGD 500 times with the uniform stochastic gradient noise for two objective functions (top and bottom). Figure (a) depicts the objective function $f$ (green, $\eta = 0$) and smoothed objectives $F$ (red and blue, $\eta > 0$). Figures (b) and (c) plot convergent points by SGD and averaged SGD with histograms, respectively.

training deep neural networks, and Izmailov et al. (2018) found the averaged SGD with not small but large step size works quite well.

The success of using a large step size can be attributed to the strong implicit bias as discussed in Izmailov et al. (2018). SGD with a large step size cannot stay in sharp regions because of the amplified stochastic gradient noise, and thus it moves to another region. After a long run, SGD will finally oscillate according to an invariant distribution covering a flat region. Then, by taking the average, we can get the mean of this distribution, which is located inside a flat region. Although this provides a good insight into how the averaged SGD with a large step size behaves, the theoretical understanding remains elusive. Hence, the research problem we aim to address is

*Why does the averaged SGD with a large step size converge to a flat region more stably than SGD?*

In our work, we address this question via the convergence analysis of both SGD and averaged SGD.

## 1.1 CONTRIBUTIONS

We first explain the idea behind our study. Our analysis builds upon the *alternative view* of SGD (Kleinberg et al., 2018) which suggested that SGD implicitly optimizes the smoothed objective function obtained by the convolution with the stochastic gradient noise (see the left of Figure 1). Since as pointed out later the smoothed objective is essentially a penalized objective on the sharpness whose strength depends on the step size, the more precise optimization of the smoothed objective with a large step size implies the convergence to a flatter region. At the same time, the step size is known to control the optimization accuracy of SGD, that is, we need to take a small step size at the final phase of training to converge. These observations indicate the *bias-optimization tradeoff* coming from the stochastic gradient noise and controlled by the step size:

*A large step size amplifies the bias towards a flat region but makes the optimization for the smoothed objective inaccurate, whereas a small step size weakens the bias but makes the optimization accurate.*

In our work, we prove that the averaged SGD can improve the above tradeoff, that is, it can optimize the smoothed objective more precisely than SGD under the same step size. Specifically, we prove as long as the smoothed objective satisfies one-point strong convexity at the solution and some regularity conditions, SGD using the step size $\eta$ converges to a distance $O(\sqrt{\eta})$ from the solution (Theorem 1), whereas the averaged SGD using the same step size converges to a distance $O(\eta)$ (Theorem 2).

We remark that *large step size* in our study means the step size with which SGD oscillates and poorly performs but the averaged SGD works. Clearly, a larger step size regardless of the condition of

the objective will diverge, thus it should be appropriately small to achieve sufficient optimization. The better dependence of $O(\eta)$ than $O(\sqrt{\eta})$ means that the averaged SGD can work well with a wider range of step sizes than SGD. Although, a too small step-size does not always bias the solution because the deviation of the solution is $O(\eta^2)$, the above dfference of the order can make the separation between SGD and averaged SGD with an appropriately chosen step-size depending on the problem. As a result, we can expect the improvement by the averaged SGD for datasets such that the stronger implicit bias with the appropriately larger step size is useful.

The separation between SGD and averaged SGD regarding the bias can occur even in the simple setup as seen in Figure 1 which depicts obtained parameters by running SGD and averaged SGD 500 times in two cases. We observe (a) both methods with the small step size can get stuck at sharp valleys or an edge of a flat region because of weak bias and accurate optimization, (b) SGD with the large step size amplifies the bias and reaches a flat region but is unstable, and (c) averaged SGD with large step size can converge stably to a near biased solution which minimizes the smoothed objective. The behavior of the averaged SGD in an asymmetric valley (the top of Figure 1), that the parameter is biased toward a flat side from an edge of the region, is also known to be preferable property in generalization as well as flat minima (see Izmailov et al. (2018); He et al. (2019)). We note that this phenomenon is certainly captured by our theory. Indeed, Figure 2 shows the convergent point of the averaged SGD is almost the minimizer of smoothed objective for each step size.

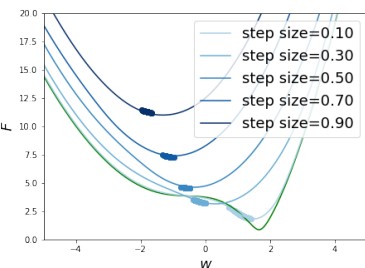

Figure 2: The figure plots the original objective (green), smoothed objectives (blue, darker is smoother), and convergent points obtained by the averaged SGD which is run for the asymmetric valley objective 500 times for each step size $\eta \in \{0, 1, 0.3, 0.5, 0.7, 0.9\}$.

Our findings are summarized below:

- SGD and averaged SGD implicitly optimize the smoothed objective, whose strength depends on the step size, up to $O(\sqrt{\eta})$ and $O(\eta)$ errors in Euclidean distance from the solution. This explains why these methods reach a flat region with an approprie step-size, since smoothing eliminates sharp minima.

- This means that averaged SGD can optimize the smoothed objective more precisely than SGD under the same step size as long as required conditions uniformly hold regarding the step size, resulting in a stronger bias towards a flat region. In other words, averaged SGD better controls the bias-optimization tradeoff than SGD.

- Hence, the parameter averaging yields an improvement for difficult datasets such that the stronger implicit bias with the larger step size is useful. This suggests the use of larger step size for such datasets so that averaged SGD stably converges but SGD itself is unstable to effectively bring out the implicit bias.

**Technical difference from Kleinberg et al. (2018).** The proof idea of Proposition 1 relies on the alternative view of SGD (Kleinberg et al., 2018) which shows the existence of an associated SGD for the smoothed objective. However, since its stochastic gradient is a biased estimator, they showed the convergence not to the solution but to a point at which a sort of one-point strong convexity holds, and avoid the treatment of a biased estimator. Hence, the optimization of the smoothed objective is not guaranteed in their theory. On the other hand, optimization accuracy is the key in our theory, thus we need nontrivial refinement of the proof under a *normal* one-point strong convexity at the solution.

## 2 PRELIMINARY – STOCHASTIC GRADIENT DESCENT

In this section, we introduce the problem setup and stochastic gradient descent (SGD) in the general form including the standard SGD for the risk minimization problems appearing in machine learning.

Let $f : \mathbb{R}^d \to \mathbb{R}$ be a smooth nonconvex objective function to be minimized. For simplicity, we assume $f$ is nonnegative. A stochastic gradient descent, randomly initialized at $w_0$, for optimizing $f$ is described as follows: for $t = 0, 1, 2, \ldots$

$$w_{t+1} = w_t - \eta \left( \nabla f(w_t) + \epsilon_{t+1}(w_t) \right), \tag{1}$$

where $\eta > 0$ is the step-size and $\epsilon_{t+1} : \mathbb{R}^d \to \mathbb{R}^d$ is a random field corresponding to the stochastic gradient noise i.e., for any $w \in \mathbb{R}^d$, $\{\epsilon_{t+1}(w)\}_{t=0}^{\infty}$ is a sequence of zero-mean random variables taking values in $\mathbb{R}^d$. A typical setup of the above is an empirical/expected risk minimization in machine learning.

**Example 1** (Risk Minimization). *Let $\ell(w, z)$ be a loss function consisting of the hypothesis function parameterized by $w \in \mathbb{R}^d$ and the data $z \in \mathbb{R}^p$. Let $\mu$ be an empirical/true data distribution over the data space and $Z$ be a random variable following $\mu$. Then, the objective function is defined by*

$$f(w) = \mathbb{E}_{Z \sim \mu}[\ell(w, Z)].$$

*Given i.i.d. random variables $\{Z_{t+1}\}_{t=0}^{\infty}$ with the same distribution as $Z$, the standard stochastic gradient at $t$-th iterate $w_t$ is defined as $\nabla_w \ell(w_t, Z_{t+1})$. In this setting, the stochastic noise $\epsilon_{t+1}$ can be $\epsilon_{t+1}(w) = \nabla_w \ell(w, Z_{t+1}) - \nabla f(w)$. Note that we can further include the $\ell_2$-regularization in the objective $f$ and the perturbation by the data augmentation in the distribution $\mu$.*

As this example satisfies, we suppose $\{\epsilon_{t+1}\}_{t=0}^{\infty}$ are independent copies each other. That is, there is a measurable map from a probability space: $\Omega \ni z \mapsto \epsilon(w, z) \in \mathbb{R}^d$, and then $\epsilon_{t+1}$ can be written as a measurable map from a product probability space: $\Omega^{\mathbb{Z}_{\geq 0}} \ni \{z_{s+1}\}_{s=0}^{\infty} \mapsto \epsilon(w, z_{t+1}) \in \mathbb{R}^d$ when explicitly representing them as measurable maps. Moreover, we make the following assumptions on the objective function and stochastic gradient noise.

**Assumption 1.**

**(A1)** $f : \mathbb{R}^d \to \mathbb{R}$ *is nonnegative, twice continuously differentiable, and its Hessian is bounded, i.e., there is a constant $L > 0$ such that for any $w \in \mathbb{R}^d$, $-LI \preceq \nabla^2 f(w) \preceq LI$.*

**(A2)** *Random fields $\{\epsilon_{t+1}\}_{t=0}^{\infty}$ are independent copies each other and each $\epsilon_{t+1}(w)$ is differentiable in $w$. Moreover, for any $w \in \mathbb{R}^d$ $\mathbb{E}[\epsilon_{t+1}(w)] = 0$ and there are $\sigma_1, \sigma_2 > 0$ such that for any $w \in \mathbb{R}^d$, $\mathbb{E}[\|\epsilon_{t+1}(w)\|^2] \leq \sigma_1^2$ and $\mathbb{E}[\|J_{\epsilon_{t+1}}^{\top}(w)\|_2] \leq \sigma_2$. , where $J_{\epsilon_{t+1}}$ is Jacobian of $\epsilon_{t+1}$.*

**Remark.** The smoothness and boundedness conditions **(A1)** on the objective function and the zero-mean and the bounded variance conditions **(A2)** on stochastic gradient noise are commonly assumed in the convergence analysis for the stochastic optimization methods. Moreover, if Hessian matrix satisfies $-LI \preceq \nabla_w^2 \ell(w, z) \preceq LI$ in Example 1, then the last condition on $J_{\epsilon_{t+1}}$ also holds with at least $\sigma_2 = 2L$ because $J_{\epsilon_{t+1}}(w) = \nabla_w^2 \ell(w, Z_{t+1}) - \nabla^2 f(w)$.

# 3 ALTERNATIVE VIEW OF STOCHASTIC GRADIENT DESCENT

An alternative view (Kleinberg et al., 2018) of SGD is the key in our analysis relating to an implicit bias towards a flat minimum. We introduce this view with a refined convergence analysis and see the bias-optimization tradeoff caused by the stochastic gradient noise with a step size.

An alternative view of SGD considers an associated iterations $\{v_t\}_{t=0}^{\infty}$ with $\{w_t\}_{t=0}^{\infty}$, which approximately minimizes a smoothed objective function obtained by the stochastic gradient noise. We here define $v_t$ as a parameter obtained by the exact gradient descent from $w_t$, that is, $v_t = w_t - \eta \nabla f(w_t)$ and we analyze the update of $v_t$ instead of $w_t$. Since $w_{t+1} = v_t - \eta \epsilon_{t+1}(w_t)$, we get $v_{t+1} = v_t - \eta \epsilon_{t+1}(w_t) - \eta \nabla f(v_t - \eta \epsilon_{t+1}(w_t))$. As shown in Appendix A. 1, under a specific setting given later, $w \mapsto v = w - \eta \nabla f(w)$ becomes a smooth invertible injection and its inverse is differentiable, thus, we identify $\epsilon_{t+1}'(v)$ with $\epsilon_{t+1}(w)$ through the map $w \mapsto v$. Then, we get an update rule of $v_t$:

$$v_{t+1} = v_t - \eta \epsilon_{t+1}'(v_t) - \eta \nabla f(v_t - \eta \epsilon_{t+1}'(v_t)). \tag{2}$$

For convenience, we refer to the rule (2) as an *implicit stochastic gradient descent* in this paper. Since, the conditional expectation of $\epsilon_{t+1}'(v_t)$ at $v_t$ is zero, we expect that the implicit SGD (2) minimizes the following smoothed objective function:

$$F(v) = \mathbb{E}[f(v - \eta \epsilon'(v))], \tag{3}$$

where $\epsilon'$ is an independent copy of $\epsilon_1', \epsilon_2', \ldots$. However, we note that this implicit SGD is not a standard SGD because $\nabla f(v_t - \eta \epsilon_{t+1}'(v_t))$ is a biased estimate of $\nabla F(v)$ (i.e., $\nabla F(v) \neq \mathbb{E}[\nabla f(v - \eta \epsilon'(v))]$) in general[1], and thus we need a detailed convergence analysis.

---

[1]By the construction, there is a probability space $(\Omega, \mathcal{F}, P)$ such that $\epsilon'(v)$ can be represented as a measurable map $\Omega \ni z \mapsto \epsilon'(v, z)$. Then, $F(v) = \mathbb{E}[f(v - \eta \epsilon'(v))] = \int f(v - \eta \epsilon'(v, z)) \mathrm{d}P(z)$ and $\nabla F(v) = \mathbb{E}[\nabla (f(v -$

The function (3) is actually a smoothed function of $f$ by the convolution using the stochastic gradient noise $\eta\epsilon'$ and the level of smoothness is controlled by the step-size $\eta$ as seen in the left of Figure 1 which depicts the original objective $f$ corresponding to $\eta = 0$ and smoothed objectives $F$. In this figure, we can observe how a nonconvex function is smoothened and its sharp local minima are eliminated by an appropriately large step size (the bottom-left figure) and how the solution is biased toward the flat side in an asymmetric valley (the top-left figure). Hence, we expect that stochastic gradient descent can avoid sharp minima and converges to a flat region. Indeed, by taking Taylor expansion of $f$, we see that $F(v)$ is an approximation of the function $f(v)$ plus the penalization on the high (positive) curvature of $f$ along the noise direction in expectation:

$$F(v) = f(v) + \frac{\eta^2}{2} \operatorname{Tr}\left(\nabla^2 f(v)\mathbb{E}[\epsilon'(v)\epsilon'(v)^\top]\right) + O(\eta^3). \tag{4}$$

The above observation indicates the reasonability of imposing some sort of convexity conditions at the solution of the smoothed objective $F(v)$ rather than the original objective $f(w)$. In this paper, we make the following one-point strong convexity at the solution $v_*$ to show the convergence of $F(v_t)$. Let $v_* = \arg\min_{v\in\mathbb{R}^d} F(v)$. We note that $F$ and $v_*$ depend on the value of $\eta$, but we do not explicitly denote this dependency for simplicity.

**Assumption 2.**

**(A3)** *There is $c > 0$ such that for any $v \in \mathbb{R}^d$, $\nabla F(v)^\top (v - v_*) \geq c\|v - v_*\|^2$.*

For instance, this assumption holds for the function in the bottom-left in Figure 1 with sufficiently large $\eta$ and for the function in the top-left figure with any $\eta$ in a certain interval $(0, \eta_0]$.

Assumption **(A3)** is a normal one-point strong convexity, whereas Kleinberg et al. (2018) assumed a different condition: $\mathbb{E}[\nabla f(v - \eta\epsilon'(v))]^\top (v - v_\circ) \geq c\|v - v_\circ\|^2$ at some parameter $v_\circ$ and showed the convergence to $v_\circ$. If $\nabla F(v) = \mathbb{E}[\nabla f(v - \eta\epsilon'(v))]$, then $v_\circ$ should be $v_*$ and both assumptions coincide. However, as noted above $\nabla F(v) \neq \mathbb{E}[\nabla f(v - \eta\epsilon'(v))]$ in general, and hence $v_\circ$ is not necessarily $v_*$. Our aim is to clarify how precisely SGD and averaged SGD can minimize $F(v)$. That is why we make the normal one-point strong convexity at $v_*$ and need a much more detailed analysis. Moreover, our proof allows for a larger step size than that in Kleinberg et al. (2018) because of the different proof techniques.

**Theorem 1.** *Under Assumption **(A1)**, **(A2)**, and **(A3)**, run SGD for $T$-iterations with the step size $\eta \leq \frac{1}{2L}$, then a sequence $\{v_t\}_{t=0}^\infty$ of the implicit SGD satisfies the following inequality:*

$$\frac{1}{T+1}\sum_{t=0}^{T} \mathbb{E}[\|v_t - v_*\|^2] \leq O\left(T^{-1}\right) + \frac{2\eta\sigma_1^2}{c} + \frac{8\eta^2\sigma_1^2 L}{3c}\left(1 + \frac{2\eta\sigma_2^2}{c}\right).$$

**Remark.** If $\sigma_1 = 0$, then SGD is nothing but deterministic gradient descent and $f = F$ because of the absence of stochastic gradient noise. Hence, SGD converges to a minimizer of $f$ according to the classical optimization theory, which is recovered by Theorem 1 with $\sigma_1 = 0$.

This theorem shows the convergence of SGD to the minimum of the smoothed objective $F$ up to distance $O(\sqrt{\eta})$ from $v_*$ as long as $F$ satisfies required assumptions even if the original objective $f$ has local minima. This is also true for $w_t$ since $\|w_t - v_t\| = O(\eta)$. Thus, convergence to a flatter region is expected through an explicit expression as a regularized objective (4). Moreover, we can see from the theorem the optimization accuracy becomes more accurate by using a smaller step size for the problem where the required conditions uniformly hold regarding $\eta$. On the other hand, a small step size clearly weakens the bias. Thus, the step size $\eta$ controls the *bias-optimization tradeoff* coming from the stochastic gradient noise.

## 4 AVERAGED SGD WITH LARGE STEP-SIZE

Izmailov et al. (2018) empirically demonstrated that averaged SGD converges to a flat region and achieves better generalization even when SGD oscillates with a relatively large step size. We

---

$\eta\epsilon'(v)))] = \int \nabla(f(v - \eta\epsilon'(v, z))) \mathrm{d}P(z)$, whereas $\nabla f(v - \eta\epsilon'(v))$ in Eq. (2) means $\nabla f(w)|_{w=v-\eta\epsilon'(v)}$ which does not involve the derivative of $\epsilon'(v)$. Therefore, $\nabla F(v) \neq \mathbb{E}[\nabla f(v - \eta\epsilon'(v))]$ in general.

theoretically attribute this phenomenon to that the averaged SGD can get closer to $v_*$ than SGD using the same step size under certain settings. In other words, parameter averaging can improve the bias-optimization tradeoff and bring out the implicit bias more effectively. In the averaged SGD, we run normal SGD (1) and take the average as follows:

$$\overline{w}_{T+1} = \frac{1}{T+1} \sum_{t=1}^{T+1} w_t.$$

Our aim is to show $\lim_{T\to\infty} \overline{w}_T$ can be closer to $v_*$ than $\{w_t\}_{t=0}^{\infty}$ and $\{v_t\}_{t=0}^{\infty}$ by clarifying the dependency of this limit on the step size $\eta$. Preferably, the implicit SGD (2) is more useful in analyzing the averaged SGD because the average $\overline{v}_T = \frac{1}{T} \sum_{t=0}^{T} v_t$ is consistent with $\overline{w}_T$ as confirmed below. By the definition, we see

$$\overline{w}_{T+1} = \overline{v}_T + \frac{1}{T+1} \sum_{t=0}^{T} \epsilon_{t+1}(w_t),$$

where the noise term $\sum_{t=0}^{T} \epsilon_{t+1}(w_t)/(T+1)$ is zero in expectation and its variance is upper bounded by $\sigma_1^2/(T+1)$ under Assumption **(A2)**. Hence, $\overline{w}_{T+1} - \overline{v}_T$ converges to zero in probability by Chebyshev's inequality; for any $r > 0$, $\mathbb{P}[\|\overline{w}_{T+1} - \overline{v}_T\| > r] \leq \sigma_1^2/(T+1)r^2 \to 0$ as $T \to \infty$, and the analysis of $\lim_{T\to\infty} \overline{w}_T$ reduces to that of $\lim_{T\to\infty} \overline{v}_T$.

We further make the additional assumptions on the smoothed objective $F : \mathbb{R}^d \to \mathbb{R}$ and give the theorem that shows the convergence of the averaged SGD.

**Assumption 3.**

**(A4)** *There is $M > 0$ such that for any $v \in \mathbb{R}^d$, $\|\nabla F(v) - \nabla^2 F(v_*)(v - v_*)\| \leq M\|v - v_*\|^2$.*

**(A5)** $\nabla^2 F(v_*)$ *is positive, i.e., there is $\mu > 0$ such that $\nabla^2 F(v_*) \succeq \mu I$.*

**Remark.** **(A4)** is used to show the superiority of the averaging scheme. This condition can be derived by the boundedness of the third-order derivative assumed in Dieuleveut et al. (2020). The positivity of Hessian **(A5)** is only required at $v_*$, which is consistent with nonconvexity. For instance, examples in Figure 1 satisfy **(A5)**.

**Theorem 2.** *Under Assumption **(A1)**–**(A5)**, run the averaged SGD for $T$-iterations with the step size $\eta \leq \frac{1}{2L}$, then the average $\overline{v}_T$ satisfies the following inequality:*

$$\|\mathbb{E}[\overline{v}_T] - v_*\| \leq O\left(T^{-\frac{1}{2}}\right) + \frac{4\sigma_1\sigma_2\eta^{\frac{3}{2}}L^{\frac{1}{2}}}{\sqrt{3}\mu} + \frac{2\eta\sigma_1^2 M}{c\mu} + \frac{8\eta^2\sigma_1^2 LM}{3c\mu}\left(1 + \frac{2\eta\sigma_2^2}{c}\right).$$

The variance of the averaged parameter $\overline{v}_T$ is typically small, hence we evaluate the distance of $\mathbb{E}[\overline{v}_T]$ to $v_*$. Indeed, this is reasonable because the central limit theorem holds for averaged SGD under the mild conditoin even for nonconvex problems (Yu et al., 2020). Theorem 2 says that the averaged SGD can optimize the smoothed objective $F$ with better accuracy of $O(\eta)$ than $O(\sqrt{\eta})$ achieved by SGD using the same step size as long as the required conditions (one-point strong convexity at minimizer and regularity for smoothed objectives) are satisfied uniformly for $\eta$ in a certain interval $(0, \eta_0]$ $(\exists \eta_0 < 1)$. These uniform requirements hold for valleys like the top-left case of Figure 1 and likely holds in the final phase of training deep neural network because of the observation that the parameter eventually falls in a better-shaped valley (see Figure 4). Therefore, we expect the averaged SGD to outperform the normal SGD in such cases, and we recommend the use of the tail-averaging scheme for deep learning as adopted in SWA (Izmailov et al., 2018), whose benefit is well known even in the convex optimization (Rakhlin et al., 2012; Mücke et al., 2019).

## 5 EXPERIMENTS

We evaluate the empirical performance of SGD and averaged SGD on image classification tasks using CIFAR10 and CIFAR100 datasets. To evaluate the usefulness of the parameter averaging for the other methods, we also compare SAM (Foret et al., 2020) with its averaging variant. We employ the tail-averaging scheme where the average is taken over the last phase of training.

Table 2: Comparison of test classification accuracies on CIFAR100 and CIFAR10 datasets.

| | | CIFAR100 | | | | CIFAR10 | | |
|---|---|---|---|---|---|---|---|---|
| | $\eta$ | ResNet-50 | WRN-28-10 | Pyramid | $\eta$ | ResNet-50 | WRN-28-10 | Pyramid |
| SGD | $s$ | 80.83 (0.21) | 81.81 (0.29) | 81.43 (0.32) | $s$ | 95.95 (0.11) | 96.85 (0.16) | 96.41 (0.22) |
| Averaged SGD | $s$ | 82.13 (0.22) | 83.13 (0.13) | 84.23 (0.03) | $s$ | 96.58 (0.14) | 97.24 (0.07) | 97.07 (0.08) |
| | $l$ | **82.87 (0.13)** | **84.23 (0.10)** | **85.12 (0.20)** | $m$ | **96.89 (0.05)** | **97.44 (0.04)** | **97.28 (0.13)** |
| SAM | $s$ | 82.56 (0.14) | 83.80 (0.27) | 84.59 (0.24) | $s$ | **96.34 (0.12)** | 97.14 (0.05) | 97.34 (0.03) |
| Averaged SAM | $s$ | 82.64 (0.12) | 84.09 (0.30) | 85.40 (0.12) | $s$ | 96.33 (0.10) | **97.21 (0.05)** | 97.34 (0.03) |
| | $l$ | **82.73 (0.28)** | **84.55 (0.17)** | **86.00 (0.04)** | $m$ | 96.31 (0.11) | 97.20 (0.06) | **97.35 (0.06)** |

We use the CNN architectures: ResNet (He et al., 2016) with 50-layers (ResNet-50), WideResNet (Zagoruyko & Komodakis, 2016) with 28 layers and width 10 (WRN-28-10), and Pyramid Network (Han et al., 2017) with 272 layers and widening factor 200. In all settings, we use the standard data augmentations: horizontal flip, normalization, padding by four pixels, random crop, and cutout (DeVries & Taylor, 2017), and we employ the weight decay with the coefficient 0.05. Moreover, we use the multi-step strategy for the step size, which decays the step size by a factor once the number of epochs reaches one of the given milestones. To

Table 1: Decay schedules for (averaged) SGD.

| $\eta$ | milestones |
|---|---|
| $s$ | {80,160,240} |
| $m$ | {80,160} |
| $l$ | {300} |

see the dependence on the step size, we use two decay schedules for the parameter averaging. Table 1 summarizes milestones labeled by the symbols: *'s'*, *'m'*, and *'l'*. The initial step size and a decay factor of the step size are set to 0.1 and 0.2 in all cases. The averages are taken from 300 epochs for the schedules *'s'* and *'l'*, and from 160 epochs for the schedule *'m'*. These hyperparameters were tuned based on the validation sets.

For a fair comparison, we run (averaged) SGD with 400 epochs and (averaged) SAM with 200 epochs because SAM requires two gradients per iteration, and thus the milestones and starting epoch of taking averages are also halved for (averaged) SAM. We evaluate each method 5 times for ResNet-50 and WRN-28-10, and 3 times for Pyramid network. The averages of classification accuracies are listed in Table 2 with the standard deviations in brackets. We observe from the table that the parameter averaging for SGD improves the classification accuracies in all cases, especially on CIFAR100 dataset. Eventually, the averaged SGD achieves comparable or better performance than SAM. Moreover, we also observe improvement by parameter averaging for SAM in most cases, which is consistent with the observations in Kaddour et al. (2022).

Comparing results on CIFAR100 and CIFAR10, the large step size is better, and the small step size is relatively poor on CIFAR100 dataset, whereas the small step size generally works on CIFAR10 dataset. If we use the step-size strategy *'l'* for CIFAR10, then the improvement becomes small (see Appendix B for this result). We hypothesize that this is because the strong bias with a large step size would be useful for difficult datasets, whereas the weak bias with a small step size would be sufficient for simple datasets such that the normal SGD already achieves high accuracies. Moreover, we note that the averaged SGD on CIFAR100 quite works well with the large step-size schedule *'l'*, but SGD itself does not converge and poorly performs under this schedule as seen in Figure 3. The accuracy of SGD temporarily increases at the 300 epochs be-

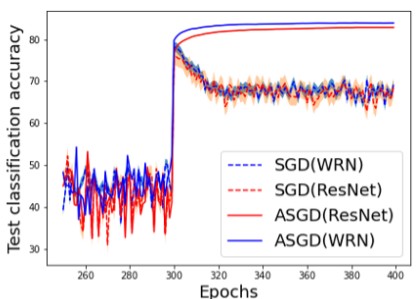

Figure 3: Test accuracies achieved by SGD and averaged SGD on CIFAR100 dataset with ResNet-50 and WRN-28-10.

cause of the decay of the step size, it decreases thereafter. However, the average of such parameters achieves significantly high accuracy as expected by our theory.

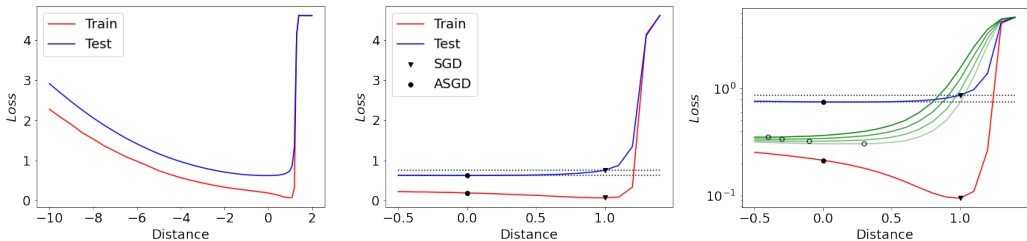

Figure 4: Sections of the train (red) and test (blue) loss landscapes across the parameters obtained by averaged SGD (distance=0) and SGD (distance=1) for ResNet-50 with CIFAR100 dataset. SGD is run with a small step size after running averaged SGD with a large step size. The middle figure is the close-up view at the edge. The triangle and circle markers represent convergent parameters by SGD and averaged SGD, respectively. The right figure plots smoothed train loss functions (green, darker is smoother) with Gaussian noises in addition to train and test losses. The blank circles are the minimizers of smoothed objectives.

Finally, we observe in Figure 4 the loss landscape around the convergent point is in better shape and forms an asymmetric valley. Therefore, we expect that the loss function around the solution uniformly satisfies the required conditions in our theory. Specifically, Figure 4 depicts the section of train and test loss functions across parameters obtained by the averaged SGD and SGD. The middle figure is the close-up view at the edge and plots each parameter. The right figure depicts the smoothed objectives with Gaussian noises in addition to train and test losses in log-scale. We observe in Figure 4 the phenomenon that SGD converges to an edge and averaged SGD converges to a flat side. This phenomenon can be explained by our theory because the minimizer of the smoothed asymptotic valley is shifted to a flat side as confirmed in a synthetic setting (Figure 2) and deep learning setting (the right of Figure 4). Moreover, the right figure indicates the possibility that the smoothed objective with appropriate stochastic gradient noise well approximates test loss, although we employ artificial noise (Gaussian) to depict graphs for simplicity. Finally, we observe that averaged SGD achieves a lower test loss which makes about 2% improvement in the classification error on CIFAR100 dataset. These observations are also consistent with the experiments conducted in He et al. (2019).

## 6 RELATED LITERATURE AND DISCUSSION

**Flat Minimum.** Keskar et al. (2017) and Hochreiter & Schmidhuber (1997) showed a flat minimum generalizes well and a sharp minimum generalizes poorly. However, the flatness solely cannot explain generalization because it can be easily manipulated (Dinh et al., 2017). Neyshabur et al. (2017) rigorously proved the sharpness combined with $\ell_2$-norm provides a generalization bound and Jiang et al. (2020) verified this correlation through large scale experiments. Keskar et al. (2017) also argued that SGD converges to a flat minimum and He et al. (2019) argued the averaged SGD tends to converge to an asymmetric valley. Several works (Kleinberg et al., 2018; Zhou et al., 2020) studied the stochastic gradient noise to theoretically prove the existence of an implicit bias towards a flat region or asymmetric valley. Moreover, many works (Izmailov et al., 2018; Foret et al., 2020; Damian et al., 2021; Orvieto et al., 2022) studied the techniques to further bring out the bias of SGD. In particular, SAM and SWA achieved a significant improvement in the generalization performance. In our paper, we show that parameter averaging stabilizes the convergence to a flat region or asymmetric valley, and suggest the usefulness of the combination with the large step size for the difficult dataset which needs a stronger regularization.

**Markov Chain Interpretation of SGD.** Dieuleveut et al. (2020); Yu et al. (2020) provided the Markov chain interpretation of SGD. They showed the marginal distribution of the parameter of SGD converges to an invariant distribution for convex and nonconvex optimization problems, respectively. Moreover, Dieuleveut et al. (2020) showed the mean of the invariant distribution, attained by the averaged SGD, is at distance $O(\eta)$ from the minimizer of the objective function, whereas SGD itself oscillates at distance $O(\sqrt{\eta})$ in the convex optimization settings. Izmailov et al. (2018) also attributed the success of SWA to such a phenomenon. That is, Izmailov et al. (2018) explained that SGD travels on the hypersphere because of the convergence to Gaussian distribution and the concentration on the

sphere under a simplified setting, and thus averaging scheme allows us to go inside of the sphere which may be flat. We can say our contribution is to theoretically justify this intuition by extending the result obtained by Dieuleveut et al. (2020) to a nonconvex optimization setting. In the proof, we utilize the alternative view of SGD (Kleinberg et al., 2018) in a non-asymptotic way under some conditions not on the original objective but on the smoothed objective function. Combination with the Markov chain view for nonconvex objective (Yu et al., 2020) may be helpful in more detailed analyses.

**Step size and Minibatch.** SGD with a large step size often suffers from stochastic gradient noise and becomes unstable. This is the reason why we should take a smaller step size so that SGD converges. In this sense, the minibatching of stochastic gradients clearly plays the same role as the step size and sometimes brings additional gains. For instance, Smith et al. (2017) empirically demonstrated that the number of parameter updates can be reduced, maintaining the learning curves on both training and test datasets by increasing minibatch size instead of decreasing step size. We remark that our analysis can incorporate the minibatch by dividing $\sigma_1^2$ and $\sigma_2^2$ in Theorem 1 and 2 by the minibatch size, and we can see certain improvements of optimization accuracy as well. Then, both SGD and averaged SGD share the same dependency on the minibatch size and thus controlling step size seems more beneficial for parameter averaging.

**Edge of Stability.** Recently, Cohen et al. (2021) showed the deterministic gradient descent for deep neural networks enters Edge of Stability phase. In the traditional optimization theory, the step size is set to be smaller than $1/L$ to ensure stable convergence and we also make such a restriction. On the other hand, the Edge of Stability phase appears when using a higher step size than $2/L$. In this phase, the training loss behaves non-monotonically and the sharpness finally stabilizes around $2/\eta$. This can be explained as follows (Lewkowycz et al., 2020); if the sharpness around the current parameter is large compared to the step size, then gradient descent cannot stay in such a region and goes to a flatter region that can accommodate the large step size. There are works (Arora et al., 2022; Ahn et al., 2022) which attempted to rigorously justify Edge of Stability phase. Interestingly, their analyses are based on a similar intuition to ours, but we consider a different regime of step sizes and a different factor (stochastic noise or larger step size than $2/L$) brings the implicit bias towards flat regions. We believe establishing a unified theory is interesting future research.

**Averaged SGD.** The averaged SGD (Ruppert, 1988; Polyak & Juditsky, 1992) is a popular variant of SGD, which returns the average of parameters obtained by SGD aiming at stabilizing the convergence. Because of the better generalization performance, many works conducted convergence rate analysis in the expected risk minimization setting and derived the asymptotically optimal rates $O(1/\sqrt{T})$ and $O(1/T)$ for non-strongly convex and strongly convex problems (Nemirovski et al., 2009; Bach & Moulines, 2011; Rakhlin et al., 2012; Lacoste-Julien et al., 2012). However, the schedule of step size is basically designed to optimize the original objective function, and hence the implicit bias coming from the large step size will eventually disappear. When applying a non-diminishing step size schedule, the non-zero optimization error basically remains. What we do in this paper is to characterize it as the implicit bias toward a flat region.

## CONCLUSION

In this paper, we showed that parameter averaging improves the bias-optimization tradeoff caused by the stochastic gradient noise. Specifically, we proved that averaged SGD optimizes the smoothed objective functions up to $O(\eta)$-error, whereas SGD itself optimizes it up to $O(\sqrt{\eta})$-error in terms of Euclidean distance from the solution, where $\eta$ is the step size. Therefore, parameter averaging significantly stabilizes the implicit bias toward a flat region, and we can expect improved performance for difficult datasets such that the stronger bias induced by a larger step size is helpful. Finally, we observed the consistency of our theory with the experiments on image classification tasks. In addition to the above discussion, another interesting research direction is to investigate what type of noise is strongly related to generalization performance.

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

# Appendix

## A   PROOFS

### A. 1   IMPLICIT STOCHASTIC GRADIENT DESCENT

Let denote by $\varphi : \mathbb{R}^d \to \mathbb{R}^d$ a change of variables from $w$ to $v$ introduced in Section 3, i.e., $v = \varphi(w) = w - \eta \nabla f(w)$.

**Lemma A.** *Under Assumption* **(A1)** *and* $\eta \leq \frac{1}{2L}$, *the function* $\varphi$ *is injective and invertible, and its inverse* $\varphi^{-1}$ *defined on on* $\mathrm{Im}\varphi$ *is differentiable.*

*Proof.* For $w, w' \in \mathbb{R}^d$, we suppose $\varphi(w) = \varphi(w')$. Then, it holds that

$$\|w - w'\| = \eta\|\nabla f(w) - \nabla f(w')\| \leq \eta L\|w - w'\| \leq \frac{1}{2}\|w - w'\|,$$

where we used $L$-Lipschitz continuity of $\nabla f$ due to **(A1)**. Therefore, we see $w = w'$ and $\varphi$ is an injection. Moreover, since $J_\varphi(w) = I - \eta \nabla^2 f(w) \succeq (1 - \eta L)I \succeq \frac{1}{2}I$. Thus, $\varphi$ is invertible and $\varphi^{-1}$, which is defined on $\mathrm{Im}\varphi$, is differentiable because of the injectivity and the inverse map theorem. $\square$

Using $\varphi$, we see $\epsilon'(v) = \epsilon(\varphi^{-1}(v))$ for $v \in \mathrm{Im}\varphi$. Let $(\Omega, \mathcal{F}, P)$ be a probability space such that $\epsilon'(v)$ can be represented as a measurable map $z \in \Omega \mapsto \epsilon'(v, z)$. Note that we use $\epsilon'(v)$ and $\epsilon'(v, z)$ depending on the situation. For a function $g : \mathbb{R}^d \to \mathbb{R}^d$, we denote by $J_g(w)$ Jacobian of $g$, i.e., $J_g(w) = (\partial g_i(w)/\partial w_j)_{i,j=1}^d$.

**Lemma B.** *Under Assumption* **(A1)** *and* **(A2)**, *we get for any* $v \in \mathrm{Im}\varphi \subset \mathbb{R}^d$,

$$\nabla F(v) = \mathbb{E}[\nabla f(v - \eta\epsilon'(v))] - \eta \int J_{\epsilon'(\cdot,z)}^\top(v)\nabla f(v - \eta\epsilon'(v,z))\mathrm{d}P(z).$$

*Moreover, if* $\eta \leq \frac{1}{2L}$, *then*

$$\|\nabla F(v) - \mathbb{E}[\nabla f(v - \eta\epsilon'(v))]\| \leq 2\eta\sigma_2\sqrt{\mathbb{E}\left[\|\nabla f(v - \eta\epsilon'(v))\|^2\right]}.$$

*Proof.* The first equality of the statement can be confirmed by the direct calculation as follows:

$$\begin{aligned}
\nabla F(v) &= \nabla\mathbb{E}[f(v - \eta\epsilon'(v))] \\
&= \int \nabla(f(v - \eta\epsilon'(v,z)))\mathrm{d}P(z) \\
&= \int (I - \eta J_{\epsilon'(\cdot,z)}^\top(v))\nabla f(v - \eta\epsilon'(v,z))\mathrm{d}P(z) \\
&= \mathbb{E}[\nabla f(v - \eta\epsilon'(v))] - \eta \int J_{\epsilon'(\cdot,z)}^\top(v)\nabla f(v - \eta\epsilon'(v,z))\mathrm{d}P(z).
\end{aligned}$$

Next, we evaluate the last term below. By the chain rule and inverse map theory,

$$J_{\epsilon'(\cdot,z)}(v) = J_{\epsilon(\phi^{-1}(\cdot),z)}(v) = J_{\epsilon(\cdot,z)}(\phi^{-1}(v))J_{\phi^{-1}}(v) = J_{\epsilon(\cdot,z)}(\varphi^{-1}(v))J_\varphi^{-1}(\varphi^{-1}(v)).$$

Note that from assumption for any $w \in \mathbb{R}^d$, $J_\varphi(w) = I - \eta\nabla^2 f(w) \succeq (1 - \eta L)I \succeq \frac{1}{2}I$. Hence,

$$\|J_{\epsilon'(\cdot,z)}^\top(v)\|_2 \leq \|J_\varphi^{-1}(\varphi^{-1}(v))\|_2\|J_{\epsilon(\cdot,z)}^\top(\varphi^{-1}(v))\|_2 \leq 2\sigma_2.$$

Finally, we get

$$\left\| \int J_{\epsilon'(\cdot,z)}^\top(v)\nabla f(v - \eta\epsilon'(v,z))\mathrm{d}P(z) \right\|$$

$$\leq \sqrt{\int \|J_{\epsilon'(\cdot,z)}^\top(v)\|_2^2 \|\nabla f(v - \eta\epsilon'(v,z))\|^2 \mathrm{d}P(z)}$$

$$\leq 2\sigma_2 \sqrt{\int \|\nabla f(v - \eta\epsilon'(v,z))\|^2 \mathrm{d}P(z)}$$

$$\leq 2\sigma_2 \sqrt{\mathbb{E}\left[\|\nabla f(v - \eta\epsilon'(v))\|^2\right]}.$$

This finishes the proof. $\qquad\square$

## A. 2 PROOF OF THEOREM 1

The following proposition is the restatement of the well-known convergence result to a stationary point using the coordinate $v$.

**Proposition A.** *Under Assumption* **(A1)**, **(A2)**, *and* $\eta \leq \frac{1}{2L}$, *we get*

$$\sum_{t=0}^{T} \mathbb{E}\left[\|\nabla f(v_t - \eta\epsilon'_{t+1}(v_t))\|^2\right] \leq \frac{4}{3\eta}\mathbb{E}[f(w_0)] + \frac{2}{3}\eta\sigma_1^2 L(T+2). \tag{5}$$

*Proof.* It is known that **(A1)** derives the following (Nesterov, 2004): for any $w, w' \in \mathbb{R}^d$,

$$f(w') \leq f(w) + \nabla f(w)^\top(w' - w) + \frac{L}{2}\|w' - w\|^2. \tag{6}$$

Substituting the update Eq. (1) into this inequality with $w' = w_{t+1}$ and $w = w_t$, and taking the conditional expectation $\mathbb{E}[\cdot|\mathcal{F}_t]$, we get

$$\mathbb{E}[f(w_{t+1})|\mathcal{F}_t] \leq f(w_t) - \eta\|\nabla f(w_t)\|^2 + \frac{\eta^2 L}{2}\mathbb{E}\left[\|\nabla f(w_t) + \epsilon_{t+1}(w_t)\|^2|\mathcal{F}_t\right]$$

$$= f(w_t) - \eta\left(1 - \frac{\eta L}{2}\right)\|\nabla f(w_t)\|^2 + \frac{\eta^2 L}{2}\mathbb{E}\left[\|\epsilon_{t+1}(w_t)\|^2|\mathcal{F}_t\right]$$

$$\leq f(w_t) - \frac{3\eta}{4}\|\nabla f(w_t)\|^2 + \frac{\eta^2\sigma_1^2 L}{2}.$$

Thus, we have $\mathbb{E}[f(w_{t+1})] \leq \mathbb{E}[f(w_t)] - \frac{3\eta}{4}\mathbb{E}[\|\nabla f(w_t)\|^2] + \frac{\eta^2\sigma_1^2 L}{2}$. By summing up this inequality, we get

$$\sum_{t=0}^{T+1} \mathbb{E}[\|\nabla f(w_t)\|^2] \leq \frac{4}{3\eta}\mathbb{E}[f(w_0)] + \frac{2}{3}\eta\sigma_1^2 L(T+2),$$

where we used the nonnegativity of $f$. By dropping the term with $t = 0$ of the sum in the left hand side and using $w_{t+1} = v_t - \eta\epsilon'_{t+1}(v_t)$, we finally get

$$\sum_{t=0}^{T} \mathbb{E}[\|\nabla f(v_t - \eta\epsilon'_{t+1}(v_t))\|^2] \leq \frac{4}{3\eta}\mathbb{E}[f(w_0)] + \frac{2}{3}\eta\sigma_1^2 L(T+2).$$

$\qquad\square$

Using the above results, we prove Theorem 1, which is restated below.

**Theorem A.** *Under Assumption* **(A1)**, **(A2)**, *and* **(A3)**, *run the stochastic gradient descent with $T$-iterations with the step size $\eta \leq \frac{1}{2L}$, then the implicit SGD satisfies the following inequality:*

$$\frac{1}{T+1}\sum_{t=0}^{T}\mathbb{E}[\|v_t - v_*\|^2] \leq \frac{1}{c\eta(T+1)}\mathbb{E}[\|v_0 - v_*\|^2] + \frac{8}{3c(T+1)}\left(1 + \frac{2\eta\sigma_2^2}{c}\right)\mathbb{E}[f(w_0)]$$

$$+ \frac{2\eta\sigma_1^2}{c} + \frac{8\eta^2\sigma_1^2 L}{3c}\left(1 + \frac{2\eta\sigma_2^2}{c}\right)$$

$$= O\left(T^{-1}\right) + \frac{2\eta\sigma_1^2}{c} + \frac{8\eta^2\sigma_1^2 L}{3c}\left(1 + \frac{2\eta\sigma_2^2}{c}\right).$$

*Proof of Theorem A.* To evaluate $\|v_{t+1} - v_*\|^2$ for the implicit SGD (2), we first give several bounds as follows. By Assumption **(A3)**, Young's inequality, and Lemma B, we get

$$-2(v_t - v_*)^\top \mathbb{E}[\nabla f(v_t - \eta\epsilon'_{t+1}(v_t))|\mathcal{F}_t]$$

$$= -2(v_t - v_*)^\top \nabla F(v_t) + 2(v_t - v_*)^\top (\nabla F(v_t) - \mathbb{E}[\nabla f(v_t - \eta\epsilon'_{t+1}(v_t))|\mathcal{F}_t])$$

$$\leq -2c\|v_t - v_*\|^2 + c\|v_t - v_*\|^2 + \frac{1}{c}\|\nabla F(v_t) - \mathbb{E}[\nabla f(v_t - \eta\epsilon'_{t+1}(v_t))|\mathcal{F}_t]\|^2$$

$$\leq -c\|v_t - v_*\|^2 + \frac{4\eta^2\sigma_2^2}{c}\mathbb{E}\left[\|\nabla f(v_t - \eta\epsilon'_{t+1}(v_t))\|^2|\mathcal{F}_t\right].$$

By Assumption **(A2)** and Young's inequality again, we get

$$\mathbb{E}[\|\epsilon'_{t+1}(v_t) + \nabla f(v_t - \eta\epsilon'_{t+1}(v_t))\|^2|\mathcal{F}_t]$$

$$\leq 2\mathbb{E}[\|\epsilon'_{t+1}(v_t)\|^2|\mathcal{F}_t] + 2\mathbb{E}[\|\nabla f(v_t - \eta\epsilon'_{t+1}(v_t))\|^2|\mathcal{F}_t]$$

$$\leq 2\sigma_1^2 + 2\mathbb{E}[\|\nabla f(v_t - \eta\epsilon'_{t+1}(v_t))\|^2|\mathcal{F}_t].$$

Combining the above two inequalities, we get

$$\mathbb{E}[\|v_{t+1} - v_*\|^2|\mathcal{F}_t] = \mathbb{E}[\|v_t - \eta\epsilon'_{t+1}(v_t) - \eta\nabla f(v_t - \eta\epsilon'_{t+1}(v_t)) - v_*\|^2|\mathcal{F}_t]$$

$$= \|v_t - v_*\|^2 - 2\eta(v_t - v_*)^\top \mathbb{E}[\nabla f(v_t - \eta\epsilon'_{t+1}(v_t))|\mathcal{F}_t]$$

$$+ \eta^2 \mathbb{E}[\|\epsilon'_{t+1}(v_t) + \nabla f(v_t - \eta\epsilon'_{t+1}(v_t))\|^2|\mathcal{F}_t]$$

$$\leq (1 - c\eta)\|v_t - v_*\|^2 + 2\eta^2\sigma_1^2$$

$$+ 2\eta^2\left(1 + \frac{2\eta\sigma_2^2}{c}\right)\mathbb{E}\left[\|\nabla f(v_t - \eta\epsilon'_{t+1}(v_t))\|^2|\mathcal{F}_t\right].$$

Taking the expectation regarding all histories and summing up over $t = 0, 1, \ldots, T$, we get

$$c\eta\sum_{t=0}^{T}\mathbb{E}[\|v_t - v_*\|^2] \leq \mathbb{E}[\|v_0 - v_*\|^2] - \mathbb{E}[\|v_{T+1} - v_*\|^2] + 2\eta^2\sigma_1^2(T+1)$$

$$+ 2\eta^2\left(1 + \frac{2\eta\sigma_2^2}{c}\right)\sum_{t=0}^{T}\mathbb{E}\left[\|\nabla f(v_t - \eta\epsilon'_{t+1}(v_t))\|^2\right]$$

$$\leq \mathbb{E}[\|v_0 - v_*\|^2] - \mathbb{E}[\|v_{T+1} - v_*\|^2] + 2\eta^2\sigma_1^2(T+1)$$

$$+ \frac{8}{3}\eta\left(1 + \frac{2\eta\sigma_2^2}{c}\right)\mathbb{E}[f(w_0)] + \frac{4}{3}\eta^3\left(1 + \frac{2\eta\sigma_2^2}{c}\right)\sigma_1^2 L(T+2),$$

where we used Proposition A. Therefore, we conclude

$$\frac{1}{T+1}\sum_{t=0}^{T}\mathbb{E}[\|v_t - v_*\|^2] \leq \frac{1}{c\eta(T+1)}\mathbb{E}[\|v_0 - v_*\|^2] + \frac{8}{3c(T+1)}\left(1 + \frac{2\eta\sigma_2^2}{c}\right)\mathbb{E}[f(w_0)]$$

$$+ \frac{2\eta\sigma_1^2}{c} + \frac{8\eta^2}{3c}\left(1 + \frac{2\eta\sigma_2^2}{c}\right)\sigma_1^2 L$$

$$= O\left(T^{-1}\right) + \frac{2\eta\sigma_1^2}{c} + \frac{8\eta^2}{3c}\left(1 + \frac{2\eta\sigma_2^2}{c}\right)\sigma_1^2 L.$$

$\square$

## A.3 Proof of Theorem 2

We give several statements used to prove Theorem 2.

**Lemma C.** *Under the same assumptions as in Theorem A, run the stochastic gradient descent with $T$-iterations with the step size $\eta \leq \frac{1}{2L}$, then the implicit SGD satisfies the following inequality:*

$$\left\| \mathbb{E}\left[ \sum_{t=0}^{T} \nabla f(v_t - \eta\epsilon'_{t+1}(v_t)) \right] \right\| \leq \frac{1}{\eta}O(1) + \frac{1}{\eta}\sqrt{(T+1)\left( \frac{4\eta\sigma_1^2}{c} + \frac{16\eta^2\sigma_1^2 L}{3c}\left(1 + \frac{2\eta\sigma_2^2}{c}\right) \right)},$$

$$\mathbb{E}\left[ \left\| \sum_{t=0}^{T} \left( \nabla F(v_t) - \mathbb{E}\left[ \nabla f(v_t - \eta\epsilon'_{t+1}(v_t))|\mathcal{F}_t \right] \right) \right\| \right] \leq 2\sigma_2 \eta^{\frac{1}{2}} O(T^{\frac{1}{2}}) + 2\sigma_1\sigma_2\eta^{\frac{3}{2}}\sqrt{\frac{2}{3}L(T+1)(T+2)}.$$

*Proof of Lemma C.* By the simple calculation, we get

$$\left\| \mathbb{E}\left[ \sum_{t=0}^{T} \nabla f(v_t - \eta\epsilon'_{t+1}(v_t)) \right] \right\| = \left\| \mathbb{E}\left[ \sum_{t=0}^{T} \left( \nabla f(v_t - \eta\epsilon'_{t+1}(v_t)) + \epsilon'_{t+1}(v_t) \right) \right] \right\|$$

$$= \frac{1}{\eta}\left\| \mathbb{E}\left[ v_0 - v_{T+1} \right] \right\|$$

$$= \frac{1}{\eta}\mathbb{E}\left[ \left\| v_0 - v_{T+1} \right\| \right]$$

$$= \frac{1}{\eta}\sqrt{\mathbb{E}\left[ \left\| v_0 - v_{T+1} \right\|^2 \right]}$$

$$\leq \frac{1}{\eta}\sqrt{2\mathbb{E}\left[ \left\| v_0 - v_* \right\|^2 + \left\| v_{T+1} - v_* \right\|^2 \right]}$$

$$\leq \frac{1}{\eta}\sqrt{2\sum_{t=0}^{T+1}\mathbb{E}\left[ \left\| v_t - v_* \right\|^2 \right]}$$

$$\leq \frac{1}{\eta}\sqrt{O(1) + (T+1)\left( \frac{4\eta\sigma_1^2}{c} + \frac{16\eta^2\sigma_1^2 L}{3c}\left(1 + \frac{2\eta\sigma_2^2}{c}\right) \right)}$$

$$\leq \frac{1}{\eta}O(1) + \frac{1}{\eta}\sqrt{(T+1)\left( \frac{4\eta\sigma_1^2}{c} + \frac{16\eta^2\sigma_1^2 L}{3c}\left(1 + \frac{2\eta\sigma_2^2}{c}\right) \right)},$$

where we used Theorem A.

Next, we show the second inequality by using Lemma B as follows:

$$\mathbb{E}\left[ \left\| \sum_{t=0}^{T} \left( \nabla F(v_t) - \mathbb{E}\left[ \nabla f(v_t - \eta\epsilon'_{t+1}(v_t))|\mathcal{F}_t \right] \right) \right\| \right]$$

$$\leq \mathbb{E}\left[ \sum_{t=0}^{T} \left\| \nabla F(v_t) - \mathbb{E}\left[ \nabla f(v_t - \eta\epsilon'_{t+1}(v_t))|\mathcal{F}_t \right] \right\| \right]$$

$$\leq \mathbb{E}\left[ \sum_{t=0}^{T} 2\eta\sigma_2\sqrt{\mathbb{E}\left[ \|\nabla f(v_t - \eta\epsilon'_{t+1}(v_t))\|^2|\mathcal{F}_t \right]} \right]$$

$$\leq 2\eta\sigma_2\sum_{t=0}^{T}\sqrt{\mathbb{E}\left[ \|\nabla f(v_t - \eta\epsilon'_{t+1}(v_t))\|^2 \right]}$$

$$\leq 2\eta\sigma_2\sqrt{(T+1)\sum_{t=0}^{T}\mathbb{E}\left[ \|\nabla f(v_t - \eta\epsilon'_{t+1}(v_t))\|^2 \right]}$$

$$\leq 2\eta\sigma_2\sqrt{\frac{1}{\eta}O(T) + \frac{2}{3}\eta\sigma_1^2 L(T+1)(T+2)}$$

$$\leq 2\sigma_2\eta^{\frac{1}{2}}O(T^{\frac{1}{2}}) + 2\sigma_1\sigma_2\eta^{\frac{3}{2}}\sqrt{\frac{2}{3}L(T+1)(T+2)}.$$

$\square$

**Proposition B.** *Under the same assumptions as in Theorem A, run the stochastic gradient descent with $T$-iterations with the step size $\eta \leq \frac{1}{2L}$, then the implicit SGD satisfies the following inequality:*

$$\frac{1}{T+1}\left\|\mathbb{E}\left[\sum_{t=0}^{T}\nabla F(v_t)\right]\right\| \leq O(T^{-\frac{1}{2}}) + \frac{4}{\sqrt{3}}\sigma_1\sigma_2\eta^{\frac{3}{2}}L^{\frac{1}{2}}.$$

*Proof of Proposition B.* Using Lemma C, we get

$$\frac{1}{T+1}\left\|\mathbb{E}\left[\sum_{t=0}^{T}\nabla F(v_t)\right]\right\| \leq \frac{1}{T+1}\left\|\mathbb{E}\left[\sum_{t=0}^{T}\left(\nabla F(v_t) - \mathbb{E}\left[\nabla f(v_t - \eta\epsilon'_{t+1}(v_t))|\mathcal{F}_t\right]\right)\right]\right\|$$

$$+ \frac{1}{T+1}\left\|\mathbb{E}\left[\sum_{t=0}^{T}\mathbb{E}\left[\nabla f(v_t - \eta\epsilon'_{t+1}(v_t))|\mathcal{F}_t\right]\right]\right\|$$

$$\leq \frac{1}{T+1}\mathbb{E}\left[\left\|\sum_{t=0}^{T}\left(\nabla F(v_t) - \mathbb{E}\left[\nabla f(v_t - \eta\epsilon'_{t+1}(v_t))|\mathcal{F}_t\right]\right)\right\|\right]$$

$$+ \frac{1}{T+1}\left\|\mathbb{E}\left[\sum_{t=0}^{T}\nabla f(v_t - \eta\epsilon'_{t+1}(v_t))\right]\right\|$$

$$\leq 2\sigma_2\eta^{\frac{1}{2}}O(T^{-\frac{1}{2}}) + 2\sigma_1\sigma_2\eta^{\frac{3}{2}}\sqrt{\frac{2}{3}L\frac{T+2}{T+1}}$$

$$+ \frac{1}{\eta}O(T^{-1}) + \frac{1}{\eta}\sqrt{\frac{1}{T+1}\left(\frac{4\eta^2\sigma_1^2}{c} + \frac{16\eta^2\sigma_1^2 L}{3c}\left(1 + \frac{2\eta\sigma_2^2}{c}\right)\right)}$$

$$\leq O(T^{-\frac{1}{2}}) + \frac{4}{\sqrt{3}}\sigma_1\sigma_2\eta^{\frac{3}{2}}L^{\frac{1}{2}}.$$

$\square$

We here prove Theorem 2 which is restated below.

**Theorem B.** *Under Assumption (A1)–(A5), run the averaged SGD for $T$-iterations with the step size $\eta \leq \frac{1}{2L}$, then the average $\overline{v}_T$ satisfies the following inequality:*

$$\|\mathbb{E}[\overline{v}_T] - v_*\| \leq O\left(T^{-\frac{1}{2}}\right) + \frac{4\sigma_1\sigma_2\eta^{\frac{3}{2}}L^{\frac{1}{2}}}{\sqrt{3}\mu} + \frac{2\eta\sigma_1^2 M}{c\mu} + \frac{8\eta^2\sigma_1^2 LM}{3c\mu}\left(1 + \frac{2\eta\sigma_2^2}{c}\right).$$

*Proof.* We define $R(v) = \nabla F(v) - \nabla^2 F(v_*)(v - v_*)$. Then, by **(A4)**, we see $\|R(v)\| \leq M\|v - v_*\|^2$. By taking average of $R(v_t)$ over $t \in \{0, 1, \dots, T\}$ and rearranging terms, we get

$$\nabla^2 F(v_*)(\overline{v}_T - v_*) = \frac{1}{T+1}\sum_{t=0}^{T}\nabla F(v_t) - \frac{1}{T+1}\sum_{t=0}^{T}R(v_t).$$

Therefore, we get

$$\mu\|\mathbb{E}[\overline{v}_T] - v_*\| \leq \|\nabla^2 F(v_*)(\mathbb{E}[\overline{v}_T] - v_*)\|$$

$$\leq \frac{1}{T+1}\left\|\mathbb{E}\left[\sum_{t=0}^{T}\nabla F(v_t)\right]\right\| + \frac{1}{T+1}\left\|\mathbb{E}\left[\sum_{t=0}^{T}R(v_t)\right]\right\|$$

$$\leq \frac{1}{T+1}\left\|\mathbb{E}\left[\sum_{t=0}^{T}\nabla F(v_t)\right]\right\| + \frac{M}{T+1}\mathbb{E}\left[\sum_{t=0}^{T}\|v_t - v_*\|^2\right].$$

The latter and former terms can be bounded by Theorem A and Proposition B. Thus, we finally get

$$\mu\|\mathbb{E}[\bar{v}_T] - v_*\| \leq O\left(T^{-\frac{1}{2}}\right) + \frac{4\sigma_1\sigma_2\eta^{\frac{3}{2}}L^{\frac{1}{2}}}{\sqrt{3}} + \frac{2\eta\sigma_1^2 M}{c} + \frac{8\eta^2\sigma_1^2 LM}{3c}\left(1 + \frac{2\eta\sigma_2^2}{c}\right).$$

□

## B ADDITIONAL EXPERIMENTS

Table 3: Comparison of test classification accuracies on CIFAR10 dataset. All methods adopt the multi-step strategy for the step size schedule.

|  |  | CIFAR10 | |
|---|---|---|---|
|  | $\eta$ | ResNet-50 | WRN-28-10 |
| SGD | $s$ | 95.95 (0.10) | 96.85 (0.16) |
| Averaged SGD | $s$ | 96.58 (0.14) | 97.24 (0.07) |
|  | $m$ | **96.89 (0.05)** | **97.44 (0.04)** |
|  | $l$ | 96.27 (0.16) | 97.05 (0.09) |

We run SGD and averaged SGD on CIFAR10 dataset with the step size strategy 'l' under the same settings as in Section 5. Table 3 lists the results including this case. We observe that the large step size 'l' does not work so well on CIFAR10 dataset compared to other schedules. We hypothesize this is because CIFAR10 is not so difficult dataset and does not require stronger bias induced by a larger step size.

We also validate the cosine annealing strategy for the step size, which is frequently used due to its excellent performance. We used the symbols 's', 'm', and 'l' for the cosine annealing depending on the last step sizes which are set to 0, 0.004, and 0.02, respectively. The parameter averaging for averaged SGD is taken over the last quarter of the training. From the table, we observe the usefulness of parameter averaging for cosine annealing schedule as well.

Table 4: Comparison of test classification accuracies on CIFAR100 and CIFAR10 datasets. All methods adopt cosine annealing for the step-size schedule.

|  |  | CIFAR100 | | |  | CIFAR10 | | |
|---|---|---|---|---|---|---|---|---|
|  | $\eta$ | ResNet-50 | WRN-28-10 | Pyramid | $\eta$ | ResNet-50 | WRN-28-10 | Pyramid |
| SGD | $s$ | 82.26 | 82.68 | 82.97 | $s$ | 96.58 | 97.00 | 96.66 |
| Averaged SGD | $s$ | **83.89** | 84.28 | 85.14 | $s$ | **97.01** | 97.28 | 97.07 |
|  | $l$ | 83.21 | **84.49** | **85.47** | $m$ | 96.86 | **97.51** | **97.32** |
| SAM | $s$ | 83.35 | 84.64 | 86.24 | $s$ | 96.40 | 96.89 | 97.61 |
| Averaged SAM | $s$ | 83.18 | 84.94 | 86.79 | $s$ | **96.56** | 97.14 | **97.55** |
|  | $l$ | **83.58** | **85.26** | **86.84** | $m$ | 96.51 | **97.19** | 97.48 |

Finally, we run SGD, SGD with a large step size, and averaged SGD to train the standard convolutional neural network on Fashion MNIST dataset to confirm how efficiently sharpness and classification accuracy can be optimized by each method. We note the large step size used for SGD is the same as that for averaged SGD. We plot the trace of Hessian $\nabla^2 f(w)$ and test loss functions in Figure 5. From this figure, we observe that the averaged SGD converges to a flatter region and achieves the highest classification accuracy on the test dataset as expected in our theory.

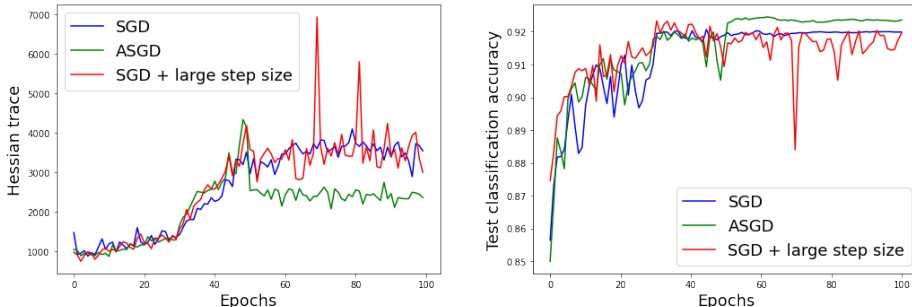

Figure 5: The figure depicts the curve of the trace of Hessian $\nabla^2 f(w)$ and test loss functions achieved by SGD, SGD with large step size, and averaged SGD. Each algorithm is run to train the standard convolutional neural network on Fashion MNIST dataset.

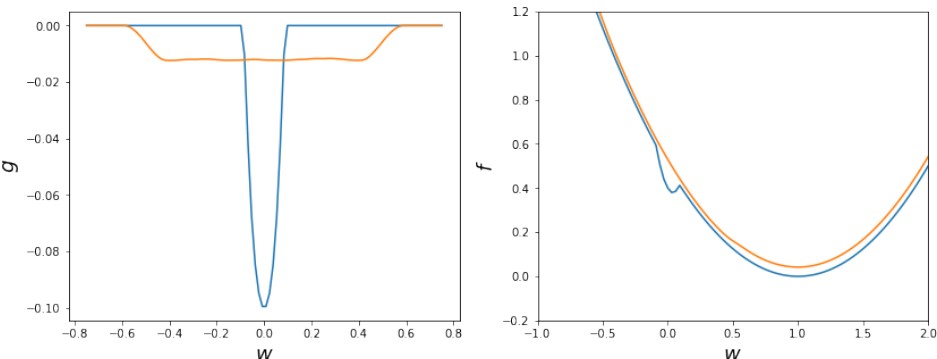

Figure 6: The left figure plots the mollifier $g_\delta$ (blue) and smoothed mollifier $G_\delta$ (orange), and the right figure plots the objective $f$ (blue) and smoothed objective $F$ (orange). The constants $\delta = 0.1, r = 2.0$, and $p = 1.0$

## C  MOTIVATING EXAMPLE

### C. 1  PROBLEM SETUP

In this section, we present a motivating example that verifies the convergence to a flat minimum and a certain separation between SGD and averaged SGD. We consider a one-dimensional objective function $f : \mathbb{R} \to \mathbb{R}$ defined below: for $p, \delta > 0$,

$$f(w) = \frac{1}{2}(w - p)^2 + g_\delta(w), \tag{7}$$

where $g_\delta : \mathbb{R} \to \mathbb{R}$ is a scaled mollifier:

$$g_\delta(w) = \begin{cases} -p\delta \exp\left(1 - \frac{1}{1-\left(\frac{w}{\delta}\right)^2}\right) & (|w| < \delta), \\ 0 & (|w| \geq \delta). \end{cases}$$

$g_\delta(w) = \delta g_1(w/\delta)$ is a scaling of the well-known mollifier of $g_1$ which is an infinitely differentiable function with a compact support. That is, $g_\delta$ is a smooth function whose support is $[-\delta, \delta]$. Because of the coefficient $p$ of $g_\delta$, the function $f(w)$ has a local minimum in $[-\delta, \delta]$, which can be the global minimum. See Figure 6 (right).

The maximum values of the first and second derivatives of $\frac{g_1}{p}$ are bounded. Thus, we define constants $C_1, C_2$ by

$$C_1 = \max\left\{1, \frac{1}{p}\max_w |g_1'(w)|\right\}, \ C_2 = \frac{1}{p}\max_w |g_1''(w)|.$$

Since $g''_\delta(w) = \frac{1}{\delta} g''_1(w/\delta)$, we see the second derivative of $g_\delta$ is bounded by $C_2 p \delta^{-1}$. Hence, Lipschitz smoothness (boundedness of Hessian) $L$ of $f$ is $1 + C_2 p \delta^{-1}$.

Next, we consider the uniform noise on the interval $[-r, r]$ for $r > 0$, i.e., $\epsilon \sim U[-r, r]$ and suppose $\epsilon(w, z) = \epsilon(z)(= \epsilon'(v, z))$ where $\Omega \ni z \mapsto \epsilon(w, z)$ is an explicit representation of the random noise. In other words, noise distribution does not change in $w$. In this case, we see $\sigma_1^2 = \mathbb{E}[\epsilon^2] \leq r^2$ and $\sigma_2 = 0$. The smoothed objective $F$ with the noise $\epsilon'$ and step-size $\eta$ is

$$F(v) = \mathbb{E}[f(v - \eta \epsilon')]$$
$$= \frac{1}{2}(v - p)^2 + \eta^2 \sigma_1^2 + \mathbb{E}[g_\delta(v - \eta \epsilon')]$$
$$\sim \frac{1}{2}(v - p)^2 + \mathbb{E}[g_\delta(v - \eta \epsilon')].$$

We consider the following problem setup:

$$\delta < \frac{p}{4(1 + 2C_1)}, \tag{8}$$

$$r > 2C_1(\delta + C_2 p). \tag{9}$$

Note that we can choose arbitrarily small $\delta > 0$ and large $r$ which satisfy the above inequalities.

For appropriate smoothing, we choose the step size $\eta$ so that

$$\frac{2C_1 \delta}{r} \leq \eta \leq \min\left\{ \frac{p}{4r} - \frac{\delta}{r}, \frac{\delta}{\delta + C_2 p} \right\}. \tag{10}$$

A step-size $\eta$ that satisfies the condition (10) exists and it also satisfies $\eta \leq 1/L = \delta/(\delta + C_2 p)$ required in the theory.

## C.2  CONVERGENCE OF SGD AND AVERAGED SGD

Under the above setup (8)–(10), we can estimate constants appearing in the convergence results of SGD and averaged SGD as follows (for the detail see the next subsection):

$$L = \frac{\delta}{\delta + C_2 p}, \; \sigma_1^2 = r^2, \; \sigma_2 = 0, \tag{11}$$

$$\mu = 1, c = \frac{1}{3}, M = \frac{8}{9p}. \tag{12}$$

Moreover, the minimum of the smoothed objective is $v_* = p$, a sharp minimum ($\sim 0$) can be eliminated by smoothing.

Therefore, for SGD we obtain by Theorem 1,

$$\frac{1}{T+1}\sum_{t=0}^{T}\mathbb{E}[\|v_t - v_*\|^2] \leq O\left(T^{-1}\right) + \frac{2\eta\sigma_1^2}{c} + \frac{8\eta^2\sigma_1^2 L}{3c}\left(1 + \frac{2\eta\sigma_2^2}{c}\right)$$

$$\leq O\left(T^{-1}\right) + 6\eta r^2 + \frac{8\eta^2 r^2 \delta}{\delta + C_2 p}\left(1 + 6\eta r^2\right).$$

We see from this inequality, $\eta_* = \frac{2C_1\delta}{r}$ is the best choice of the step-size, resulting in

$$\frac{1}{T+1}\sum_{t=0}^{T}\mathbb{E}[\|v_t - v_*\|] \leq O\left(T^{-1/2}\right) + \sqrt{12C_1\delta r + \frac{32C_1^2\delta^3}{\delta + C_2 p}\left(1 + 12C_1\delta r\right)},$$

where we apply Jensen's inequality to derive the bound on $L_1$-norm. This result means SGD avoids a sharp minimum (i.e., $v \sim 0$ under small $\delta > 0$) and converges to a flat minimum $v_* = p$, and a too large noise will affect the convergence to $v_*$ based on our step-size policy.

Moreover, for averaged SGD we obtain by Theorem 2,

$$\|\mathbb{E}[\bar{v}_T] - v_*\| \leq O\left(T^{-\frac{1}{2}}\right) + \frac{4\sigma_1\sigma_2\eta^{\frac{3}{2}}L^{\frac{1}{2}}}{\sqrt{3}\mu} + \frac{2\eta\sigma_1^2 M}{c\mu} + \frac{8\eta^2\sigma_1^2 LM}{3c\mu}\left(1 + \frac{2\eta\sigma_2^2}{c}\right)$$

$$= O\left(T^{-\frac{1}{2}}\right) + \frac{16}{9p}\left(3\eta r^2 + \frac{4\eta^2 r^2 \delta}{\delta + C_2 p}\left(1 + 6\eta r^2\right)\right)$$

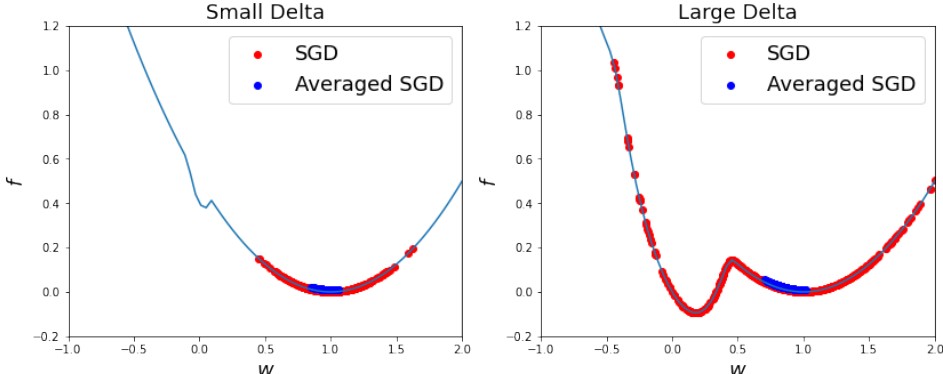

Figure 7: The figures plot the convergent points of SGD and averaged SGD for problems with $\delta = 0.1$ and $\delta = 0.5$.

Hence, for $\eta_* = \frac{2C_1\delta}{r}$ we obtain

$$\|\mathbb{E}[\overline{v}_T] - v_*\| \leq O\left(T^{-\frac{1}{2}}\right) + \frac{32}{9p}\left(3C_1\delta r + \frac{8C_1^2\delta^3}{\delta + C_2 p}\left(1 + 12C_1\delta r\right)\right).$$

This bound means averaged SGD will get closer to $v_* = p$ as long as SGD approaches a neighborhood of $v_*$

According to the above results, both SGD and averaged SGD converge to a flat region when $\delta$ is small, and averaged SGD converges even when $\delta$ is relatively large.

We empirically observed this phenomenon in Figure 7 in which we run SGD and averaged SGD for problems with small $\delta = 0.1$ and relatively large $\delta = 0.5$.

## C.3 ESTIMATION OF CONSTANTS

We verify the estimations of constants in (11). $L$, $\sigma_1^2$, and $\sigma_2$ are already obtained, thus, $mu, c$, and $M$ remain.

**Minimum and estimation of $\mu$.** We first see that under our problem setting, the local minimum around the origin is eliminated and $p$ is the optimal solution of $F$, i.e., $v_* = p$.

The smoothed function $G_\delta(v) \stackrel{\text{def}}{=} \mathbb{E}[g_\delta(v - \eta\epsilon')]$ and its derivative $G'_\delta(v)$ are calculated as follows:

$$G_\delta(v) = \int_{-r}^{r} g_\delta(v - \eta t)\frac{1}{2r}\mathrm{d}t,$$

$$G'_\delta(v) = \int_{-r}^{r} g'_\delta(v - \eta t)\frac{1}{2r}\mathrm{d}t.$$

By taking into account $\mathrm{supp}(g_\delta) = [-\delta, \delta]$, the smoothed objective $G_\delta(v)$ is constant on $\{|v| \leq \eta r - \delta\} \cup \{|v| \geq \eta r + \delta\}$, and thus, $G'_\delta$ is non-zero only on $\mathrm{supp}(G'_\delta) = [-\eta r - \delta, -\eta r + \delta] \cup [\eta r - \delta, \eta r + \delta]$. See Figure 6 (left). Since $\eta r + \delta < p/4 < p$ under (10), $v = p$ is still a local minimum of $F$.

We evaluate the bound on $G'_\delta$ on $\mathrm{supp}(G'_\delta)$ below. for $v \in [\eta r - \delta, \eta r + \delta]$ the support of $g'_\delta(v - \eta t)$ in $t \in \mathbb{R}$ is $[(v - \delta)/\eta, (v + \delta)/\eta]$, we get

$$0 \leq G'_\delta(v) = \int_{-r}^{r} g'_\delta(v - \eta t)\frac{1}{2r}\mathrm{d}t$$

$$\leq \int_{\frac{v-\delta}{\eta}}^{\frac{v+\delta}{\eta}} |g'_\delta(v - \eta t)|\frac{1}{2r}\mathrm{d}t$$

$$\leq pC_1 \int_{\frac{v-\delta}{\eta}}^{\frac{v+\delta}{\eta}} \frac{1}{2r}\mathrm{d}t = \frac{pC_1\delta}{\eta r},$$

where we used $|g'_\delta(v)| = |g'_1(v/\delta)| \le pC_1$. A bound on $[-\eta r - \delta, -\eta r + \delta]$ is also obtained in the same way. Thus, we see

$$\begin{cases} -\frac{pC_1\delta}{\eta r} \le G'_\delta(v) \le 0 & (v \in [-\eta r - \delta, -\eta r + \delta]), \\ 0 \le G'_\delta(v) \le \frac{pC_1\delta}{\eta r} & (v \in [\eta r - \delta, \eta r + \delta]), \\ G'_\delta(v) = 0 & (\text{else}). \end{cases}$$

If there are additional stationary points of $F$, they should exist in $[\eta r - \delta, \eta r + \delta] = \text{supp}(G'_\delta) \setminus [-\eta r - \delta, -\eta r + \delta]$ because of the sign of $G'_\delta$ and $\text{supp}(G'_\delta) \subset (-\infty, p/4)$. However, since $\eta r + \delta \le p/4$ and $\frac{pC_1\delta}{\eta r} \le p/2$ under (10), we see

$$\max_{v \in [\eta r - \delta, \eta r + \delta]} F'(v) \le (\eta r + \delta) - p + \frac{pC_1\delta}{\eta r} \le \frac{p}{4} - p + \frac{p}{2} = -\frac{p}{4}.$$

Hence, $v_* = p$ is the unique local minimum (i.e., optimal solution) of $F$ and we can conclude $\mu = 1$.

**Estimation of $c$.** From the above argument, we get

$$F'(v)(v - p) = (v - p)^2 + G'_\delta(v)(v - p)$$
$$\ge \begin{cases} (v - p)^2 & (v \in [-\eta r - \delta, -\eta r + \delta]), \\ (v - p)^2 + \frac{pC_1\delta}{\eta r}(v - p) \ge (v - p)^2 + \frac{p}{2}(v - p) & (v \in [\eta r - \delta, \eta r + \delta]), \\ (v - p)^2 & (\text{else}). \end{cases}$$

Clearly, $p/2 \le 2(p - v)/3$ for $v \le \eta r + \delta \le p/4$. Thus, $F'(v)(v - p) \ge (v - p)^2/3$ on $v \in [\eta r - \delta, \eta r + \delta]$ and we conclude $c = 1/3$.

**Estimation of $M$.** Noting $v_* = p$ and $F''(p) = 1$, we have

$$|F'(v) - F''(v_*)(v - v_*)| = |(v - p) + G'_\delta(v) - (v - p)| = |G'_\delta(v)|.$$

Because of the problem setup, it is enough to verify $M = \frac{8}{9p}$ satisfies $|G'_\delta(v)| \le M|v - p|^2$ on $v \in [\eta r - \delta, \eta r + \delta]$. Since $|G'_\delta(v)| \le p/2$ and $v \le p/4$ for $v$ in this interval, we have

$$|G'_\delta(v)| \le \frac{p}{2} \le M\frac{9p^2}{16} \le M(v - p)^2.$$

This concludes $M = \frac{8}{9p}$.

