# OpenReview forum: "Parameter Averaging for SGD Stabilizes the Implicit Bias towards Flat Regions"
_ICLR.cc/2023/Conference — Submitted to ICLR 2023_

### Official Review · Reviewer_vTYE · 2022-10-21

**Confidence:** 3
**Clarity, Quality, Novelty And Reproducibility:** This paper is well-written and novel.
**Correctness:** 4
**Technical Novelty And Significance:** 3
**Empirical Novelty And Significance:** 3
**Recommendation:** 8

**Strength And Weaknesses:**

Strength: This paper is well written, and it is easy to follow. The theoretical results are novel and the discussions/remarks are very helpful. While the results seem incremental over some existing work on the implicit bias of SGD, there is still a significant contribution, showing an improved rate using averaged SGD.

Weakness: The author mentioned before Theorem 1, "different proof techniques" than (Kleinber et al.), maybe you can elaborate?

**Summary Of The Paper:**

This paper presents the convergence analysis for SGD and averaged SGD with fixed step size, showing that iterates converge to the vicinity of the optimizer of a smoothed loss function. Compared to prior work, the author improved existing results with a less strict assumption on the step size, and provide a convergence result for the averaged SGD, showing a better rate over vanilla SGD.

**Summary Of The Review:**

Good paper studies a very relevant topic in the theory of machine learning.

---

> ### Author Response · Authors · 2022-11-18
> **To Reviewer vTYE**
>
> Thank you for the positive feedback.
>
> > The author mentioned before Theorem 1, "different proof techniques" than (Kleinber et al.), maybe you can elaborate?
>
> This is a good question. [Kleinber et al.] shows the convergence of $\||v_T - v_*||^2$ for the last iteration $v_T$ of SGD. It is well known that the last iteration usually oscillates because of the stochastic gradient noise, which deteriorates the convergence rate. On the other hand, instead of $\|| v_T-v_* \||^2 $, we evaluate the average distance $\frac{1}{T}\sum_{t=0}^T \|| v_t - v_* \||^2$ which is much more stable to converge. This difference gives an improvement in convergence analysis.

---

### Official Review · Reviewer_fQ6K · 2022-10-22

**Confidence:** 3
**Clarity, Quality, Novelty And Reproducibility:** check comments above
**Correctness:** 2
**Technical Novelty And Significance:** 2
**Empirical Novelty And Significance:** 2
**Recommendation:** 5

**Strength And Weaknesses:**

## Strength
* The paper is well-structured and well-written (in high quality), with a clear related work section. The reviewer was enjoying reading most part of the manuscript.
* The studied question is of great importance to the optimization community.
* Extensive numerical results are included in the manuscript in terms of various neural architectures and learning rate schedules; besides, some visualization results are very insightful.

## Weaknesses
1. The manuscript argues that SGD using the step size $\eta$ converges to a distance $O(\sqrt{\eta})$ from the solution, whereas the averaged SGD using the same step size converges to a distance to $O(\eta)$. First of all, the reviewer is unclear on how to conclude the $O(\sqrt{\eta})$ and $O(\eta)$ from Theorem 1 and Theorem 2; and authors are required to elaborate on them. More importantly, compared to Theorem 1, Theorem 2 relies on the additional Assumption 3, and thus could derive a tighter rate; however, it does not make much sense to compare two results under different assumptions, and even highlight it as a key contribution.
2. If the reviewer understands the manuscript correctly, the current convergence theory in the submission cannot capture the training dynamic caused by using a larger step size as the averaged SGD only runs normal SGD and takes the average in the end. Understanding this would be a key step in interpreting the implicit bias in SGD, however, the current draft cannot fill this gap.
3. The choice of hyper-parameters used in the manuscript seems a bit random and no justification can be found. For example, weight decay with a coefficient of 0.05 was used but the reviewer believes many SOTA training recipes use 1e-4 or 5e-4; Similar comment can be applied to the choice of decay factor as well as the milestones for three cases.

**Summary Of The Paper:**

The manuscript studies the optimization and generalization trade-off caused by stochastic noise, step size, implicit bias, and sharpness of the minimum; and more importantly, why an averaged SGD with a large step size can improve the trade-offs and converge to a flat region more stably than SGD.

**Summary Of The Review:**

In general, the manuscript is well-written, and the main concerns of the reviewer are
* unfair comparison of theoretical/numerical results
* unsupported experimental setups.

---

> ### Author Response · Authors · 2022-11-18
> **To Reviewer fQ6K**
>
> Thank you for your review and helpful feedback.
>
> > More importantly, compared to Theorem 1, Theorem 2 relies on the additional Assumption 3, and thus could derive a tighter rate; however, it does not make much sense to compare two results under different assumptions, and even highlight it as a key contribution.
>
> As you said, Theorem 2 uses additional assumptions A4 and A5. However, the same separation of the results appears in the case of the quadratic function case in which A4 and A5 are deactivated. Indeed, to justify this difference perfectly, we need to provide a lower bound on SGD even under the same assumptions. However, the lower bound analysis is much more challenging than the upper bound case, and hence many studies basically compare the performance of methods based on their upper bound in the optimization contexts. Moreover, we would like to note that similar assumptions were made by [Dieuleveut, et al. (2020)] for studying SGD and averaged SGD for convex optimization problems. Thus, we think our assumptions are also reasonable and our theory could also support the preferable ability of average SGD and SWA.
>
>
> Aymeric Dieuleveut, Alain Durmus, and Francis Bach. Bridging the gap between constant step size stochastic gradient descent and markov chains. Annals of Statistics, 2020.
>
> > If the reviewer understands the manuscript correctly, the current convergence theory in the submission cannot capture the training dynamic caused by using a larger step size as the averaged SGD only runs normal SGD and takes the average in the end.
>
> Indeed, we ran averaged SGD in the final phase of training for deep learning problems. This is because the required assumptions in our theory do not hold over the entire parameter space and hence we need a warm-start for averaged SGD to bring out its performance suggested by our theory. On the other hand, we ran averaged SGD from the beginning of training for synthetic problems and observed a stable convergence to a flat region. Especially, an additional example (Appendix C) in the revision is theoretically well supported. Please see another post. For this example, we estimate all constants which define the objective function and justify a meaning separation between SGD and averaged SGD, which can be a sanity check of our theory.
>
> > The choice of hyper-parameters used in the manuscript seems a bit random and no justification can be found. For example, weight decay with a coefficient of 0.05 was used but the reviewer believes many SOTA training recipes use 1e-4 or 5e-4; Similar comment can be applied to the choice of decay factor as well as the milestones for three cases.
>
> The coefficient of weight decay was $\lambda=0.005$. The value of 0.05 was a typo. Thank you for pointing this out. In our preliminary experiments, we also validated $\lambda=5e-4$. In our observations, the coefficient $\lambda=5e-4$ works well for SGD with the momentum method, but $\lambda=0.005$ was better for vanilla SGD. We choose the step-size schedule and $\lambda$ based on the validation for vanilla SGD. We note that the decay factor and milestones were adopted while referencing the implementation of the adaptive SAM:
>
> (Adaptive) SAM Optimize
>
> [https://github.com/davda54/sam](https://github.com/davda54/sam)

---

### Official Review · Reviewer_Q2SC · 2022-10-26

**Confidence:** 4
**Correctness:** 3
**Technical Novelty And Significance:** 3
**Empirical Novelty And Significance:** Not applicable
**Recommendation:** 5

**Clarity, Quality, Novelty And Reproducibility:**

This paper is well-written and easy to follow. The assumptions and theorems are stated very clearly.

The convergence analysis for SGD in the alternative view in this paper is new and the result is stronger than Kleinberg et al. (2018).

**Strength And Weaknesses:**

### Strength

1. The phenomenon that averaged SGD leads to a more stable convergence to flat regions is interesting to study, and may deepen the current understanding of the training dynamics in the late training phase.
2. Clear convergence results for both SGD and averaged SGD.
3. The theoretical result indeed provides a clear separation between SGD and averaged SGD in terms of the distance upper bounds to the minimizer of the smoothed loss.


### Weakness

**Major Concern.** The major concern I have is whether the current result indeed shows that larger LR leads to a flatter minimizer. If not, then people may want to use SGD with a smaller LR so that the distance to the minimizer is smaller, and then SGD may beat averaged SGD. Basically, the authors should check whether the minimizer $v_*(\eta)$ as a function of $\eta$ can move a lot when $\eta$ goes from a big value $\eta$ to a small value $\eta'$.

However, the distance between $v_*(\eta)$ and $v_*(\eta')$ should be bounded by $O(\eta^2)$. The reason is that we can apply the implicit function theorem for solving the equation $\nabla F(v) = 0$, which gives us a curve $v_*(\eta)$ wrt $\eta$. Note that the derivative wrt $\eta$ is $-(\nabla^2 F(v_*))^{-1} \left(\frac{\eta^2}{2} \nabla \mathrm{Tr}(\nabla^2 f(v_*) \mathbb{E}[\epsilon' \epsilon'^\top]) + O(\eta^3)\right)$. According to A4 and A5, this should be of order $O(\eta^2)$.

Note that the theorem for SGD shows a distance bound $O(\eta)$. So for SGD with a much smaller LR $\eta'$, the distance from the final parameter to the minimizer of the smoothed loss defined with $\eta$ would be $O(\eta^2) + O(\sqrt{\eta'}) = O(\eta^2)$. But for averaged SGD with LR $\eta$, it is $O(\eta)$, which is actually much higher.

The above argument would suggest that averaged SGD with large LR does not encourage flatness at all.

**Minor concerns.**
1. I understand that one-point convexity makes the analysis possible, but the authors should give some concrete examples of one-point convex functions, and discuss how general this notion is. It should also be stated clearly whether one-point convexity holds in the experiments or not.
2. Comparing the distance upper bounds for SGD and averaged SGD is reasonable, but this does not mean SGD cannot get much closer to the minimizer because the bound may not be tight. I would recommend the authors prove a lower bound in the SGD case to make the separation more concrete.

**Summary Of The Paper:**

This paper aims to understand how averaging the parameters of SGD (averaged SGD) leads to a more stable convergence to flat regions.
The authors take the alternative view of SGD from Kleinberg et al. (2018), which tracks $v = w - \eta \nabla f(v)$ instead of $w$ itself. Then SGD wrt $w$ can be viewed as a stochastic semi-gradient method wrt $v$ on a smoothed loss, where the smoothening factor depends on eta. Under one-point convexity at the minimizer of the smoothed loss (and also various regularity conditions), they show that both SGD and averaged SGD oscillate around the minimizer, but the distance to the minimizer from the final parameter of averaged SGD is smaller than that from SGD ($O(\eta)$ v.s. $O(\sqrt{\eta})$).

**Summary Of The Review:**

I would like to hear the authors' comments on my major concern because I could be wrong. If $v_*$ can indeed change a lot for different learning rates, I would be rather happy to raise my score. Otherwise, I have to vote for rejection, but I encourage the authors to come up with another argument to motivate their convergence analysis and resubmit.

=================

After reading the rebuttal, I found the 1-dim example given by the author helpful. This simple example shows that the theory indeed predicts the averaged SGD converges to a flatter solution than SGD, so I have raised my score from 3 to 5.

Overall, I'm not against accepting this paper because the convergence analysis still looks good to me. But I feel that the paper is at the borderline due to the following concerns:
1. As I pointed out in the initial review, this paper compares the upper bounds for SGD and averaged SGD (even in the 1-dim example, only the upper bounds are compared). A more solid theoretical result should give a lower bound for SGD to ensure that it doesn't converge to flat minimizes.
2. Though I appreciate the 1-dim example for its simplicity, it is unclear how general it is. It would be better if the authors could have generalized the example to a broad function class. Alternatively, the authors could also make up the generality issue in theory by measuring the empirical loss landscape to see if it is similar to the 1-dim example in some sense. For this reason, I believe that this paper still has much room for improvement.

---

> ### Author Response · Authors · 2022-11-18
> **To Reviewer Q2SC**
>
> Thanks for your thorough review and constructive comments.
>
> > Basically, the authors should check whether the minimizer $v_*(\eta)$ as a function of $\eta$ can move a lot when $\eta$ goes from a big value $\eta$ to a small value $\eta’$.
>
> This comment is quite important and should be addressed in the paper. Indeed, the distance between $v_*(eta)$ and $v_*(0)$ for each minimum is $O(\eta^2)$ when $\eta$ is small, as you pointed out. Thus, to make a certain separation between two solutions, we need to use a certain large step-size depending on the function. And finally, we could find a simple example that makes a meaningful separation and our theory works quite well. A similar comment was given by another reviewer, thus we wrote the detail in another post.
>
> We believe this example addressed your concern. We would like to thank the reviewer again. Your comment is really helpful and makes our paper stronger.
>
> >  It should also be stated clearly whether one-point convexity holds in the experiments or not.
>
> Synthetic examples in the paper satisfy one-point convexity. Indeed, it can be satisfied as long as the gradient of smoothed objective does not degenerate. But we did not provide specific constants because the estimation of constants was complicated. Now, we provide all constants appearing in theorems for an additional example. See another post and Appendix C.

---

### Official Review · Reviewer_EP51 · 2022-10-30

**Confidence:** 3
**Correctness:** 3
**Technical Novelty And Significance:** 3
**Empirical Novelty And Significance:** 2
**Recommendation:** 5

**Clarity, Quality, Novelty And Reproducibility:**

The paper is well-written and easy to understand. The proofs of the main theorems (Theorem 1 and 2) seem correct. The adoption of the framework (Kleinberg et al., 2018) framework toward understanding the implicit bias of SGD and SWA is quite novel. More details about experiments are needed.

**Strength And Weaknesses:**

##Pros:

1. The paper is well-written and easy to understand. The proofs of the main theorems (Theorem 1 and 2) seem correct to me.

2. The framework (Kleinberg et al., 2018) is originally developed for understanding optimization under assumptions weaker than convexity, that is, one-point convexity. The adoption of their framework towards understanding the implicit bias of SGD and SWA is quite novel, as well as the bias-optimization tradeoff view.

3. The authors successfully explain the efficacy of SWA using the bias-optimization tradeoff, that is, though SWA implicitly penalizes the same objective as SGD does, SWA optimizes the objective better than SGD.



##Cons:

1. Though the theory proposed in this paper looks self-consistent, the authors don't really justify whether their theory is relevant to practice, like either directly testing their proposed regularizer using other optimizers so there is no optimization issue, or developing algorithms with better generalization using their insight. Technically speaking, it is not clear to me whether there exists a suitable learning rate, such that the optimization-bias tradeoff gives non-trivial improvement compared to the empirical minimizer of the original training loss.

	To be more specific, under Assumption 2, it is fairly easy to see any stationary point $w^*$ of the original loss $f$ satisfies that $\eta^2 \nabla_{w^*} {Tr}(\nabla^2 f(w^*) \Sigma(w^*)) \ge c\|v^*-w^* \|$ , where $\Sigma(w^*)$ is the covariance matrix of noise in Eq. (4). By Assumption (3), $\nabla^3 f(w^*)$ is of constant magnitude, which implies that $\|w^*-v^* \| = O(\eta^2)$.  Immediately we see the optimization guarantee given by both Theorem 1 and 2 are vacuous in terms of the implicit bias. This is because we know even getting $O(\eta^2)$-close (like $w^*$) to the minimizer of the regularized loss $v^*$ is not enough ($w^*$ can be understood by the solution found SGD with tiny LR)

	That being said, I don't think the view presented in this paper is wrong. It is still valuable. It could be analysis are not tight or assumptions are too strong. But I do think the authors should address the above issue before publishing this work, which can be viewed as a sanity check to the theory.

2. Regarding the relevance to the deep learning practice, another thing I think the authors should reconcile is why in practice typically the learning rate of SGD needs to be annealed and the generalization is still good.

3. The details of the setting for experiments are missing, e.g. Figure 4. Why does ASGD have higher training loss than SGD in Figure 4? Does the same thing happen for Figure 3 (training loss is not reported there) Intuitively by averaging you should get a smaller training loss, e.g., think about SGD on a quadratic loss with isotropic gaussian noise.

**Typos**:

1.page 17. "here we prove Theorem 1 which ..". should be Theorem 2.
2.page 17. All $\nabla R$ should be $R$.


**Summary Of The Paper:**

This paper aims to explain why stochastic weight averaging (SWA, Izmailov et al., 2018) helps generalization. The paper adopts the framework of Kleinberg et al., 2018 and shows that SGD implicitly minimizes a regularized objective, where the regularizer depends on both Hessian of the loss and the covariance matrix of the stochastic gradient, under the assumption that the regularized loss is one-point convex. The authors show that the regularization strength is proportional to $O(\eta^2)$ where $\eta$ is the learning rate and SGD can reach a solution which is $O(\sqrt{\eta})$-close to the minimizer of the regularizers loss. The authors further explain the benefit of stochastic weight averaging by showing that the averaged iterate better optimizes the regularized loss, that is, SWA can get $O(\eta)$-close to the minimizer.

**Summary Of The Review:**

This paper gives a novel understanding of the implicit bias of SGD, which explains the efficacy of stochastic weight averaging. However, I am not sure how this understanding is related to practice as the optimization guarantee presented in the main theorems is quite weak, not enough to separate SGD and SWA from ERM even. Thus now I tend to reject this paper, but I am willing to increase the score if the authors can address my concerns satisfactorily.
(A good example is Damian et al.,2021, where they showed that the minimizer of the original loss doesn't meet their guarantee for label noise SGD on the regularized loss)

- Damian, Alex, Tengyu Ma, and Jason D. Lee. "Label noise sgd provably prefers flat global minimizers." Advances in Neural Information Processing Systems 34 (2021): 27449-27461.

================================================

Update after rebuttal: I've read the revision and rebuttal by the authors as well as reviews by other reviewers. My concern that "whether there exists a suitable learning rate, such that the optimization-bias tradeoff gives non-trivial improvement compared to the empirical minimizer of the original training loss" is not well-addressed by the authors and I decide to keep my score.

The new example in appendix C given by the authors in the revision shows when $\delta$ is sufficiently small, SGD and average SGD will converge to the flat local minimizer (which is also the global minimizer of original loss $f$). This is just an optimization result because any algorithm that finds global minimizer will also find the flat minimizer in this case. To make it a result of the implicit bias of SGD, the author should demonstrate there is a case where the sharp local minimizer is indeed a global minimizer but still gradient noise biases SGD to the flatter area, even if the loss there could be slightly higher.

---

> ### Author Response · Authors · 2022-11-18
> **To Reviewer EP51**
>
> Thanks for your thorough review and helpful comments.
>
> > Technically speaking, it is not clear to me whether there exists a suitable learning rate, such that the optimization-bias tradeoff gives non-trivial improvement compared to the empirical minimizer of the original training loss.
>
> This comment is very helpful and gives us an opportunity to think about a specific example that makes a certain separation between two minima. Finally, we could find a simple problem setup that verifies such a separation and our theory. For the detail, please see another post. We believe this example could address your concern.
>
>
> > Regarding the relevance to the deep learning practice, another thing I think the authors should reconcile is why in practice typically the learning rate of SGD needs to be annealed and the generalization is still good.
>
> Indeed, an annealed learning rate leads to good generalization. But, we observed a relatively high constant learning rate with averaging indeed improves the generalization performance. This is quite reproducible as pointed out in several studies. Thus, we believe the analysis of this strategy would be important.
>
>
> > The details of the setting for experiments are missing, e.g. Figure 4. Why does ASGD have higher training loss than SGD in Figure 4? Does the samething happen for Figure 3 (training loss is not reported there) Intuitively by averaging you should get a smaller training loss, e.g., think about SGDon a quadratic loss with isotropic gaussian noise.
>
>
> As you said, averaged SGD basically achieves lower training loss over SGD as well as test error. A convergence point of SGD in Figure 4 is obtained by running SGD with a small step-size after running averaged SGD with a high constant step-size, which is noted in the revision. As a result, SGD finally converges to a sharp minimum and achieves a smaller training loss. Importantly, this final convergence deteriorates a test error as reported in this Figure. This kind of convergence behavior was also reported in the following paper, and we could reproduce it.
>
> Haowei He, Gao Huang, and Yang Yuan. Asymmetric Valleys: Beyond Sharp and Flat Local Minima, NeurIPS 2019.

---

### Author Response · Authors · 2022-11-18
**To all reviewers.**

We thank all reviewers for their helpful feedback.

In particular, comments on the distance between solutions of the objective $f$ and smoothed objective $F$ are very important to be addressed. I really appreciate Reviewers EP51 and Q2SC. Indeed, we also confirmed that the distance between these two solutions is $O(\eta^2)$. Hence, we need to consider a certain step-size depending on the function to make a meaningful separation. Finally, we could find such a simple example by estimating all constants appearing in our theorems. For the detail see Appendix C.

This example is the sum of a flat quadratic function $(x-p)^2$ and a sharp mollifier $g_\delta$ centered at $0$. Thus, there are two flat and sharp local minima and their distance is $p=O(1)$. Each minimum could be the global minimum. The constants which define the shape of $f$ and $F$ are the sharpness (radius) $\delta$ of the mollifier $g_\delta$ (smaller $\delta$ is sharper) and the radius of the uniform noise $r$.

For this problem, we can find a certain condition on $\delta, r$, and $\eta$ so that the sharp minimum is eliminated and converges to a flat solution $p$. Specifically, we show SGD and averaged SGD with the step-size satisfying $\delta/r \lesssim \eta \lesssim \min\\{1/r, \delta\\}$ can avoid a sharp minimum and converge to an arbitrarily small neighborhood of $p$ when $\delta$ is small even though the distance between two minimum is $p=O(1)$. For the detailed conditions, see Eq. (8)-(10) in Appendix C. Moreover, the superior convergence of averaged SGD over SGD is also verified by our theory in this example.

I think this is the first result that certainly shows the ability of SGD and averaged SGD to avoid a sharp minimum and converge to a flat minimum based on the alternative view technique. We would appreciate it if the reviewers could update your evaluation in light of our response.

The major changes are summarized below. These changes are colored red in the revision.

- We added an example to Appendix C which verifies our theory (see above).
- We noted that the distance between $f$ and $F$ is $O(\eta^2)$, and thus, an appropriate step-size should be chosen to make a certain separation.
- We relaxed Assumption (A4) which originally supposed three times continuous differentiability. However, a looser condition was used in the proof, thus, we rewrite this assumption.
- We improved Theorem 2 by studying $\mathbb{E}[v_T]$ instead of $v_T$ for averaged SGD . The proof is almost the same but a little simplified.

---

> ### Author Response · Authors · 2022-11-22
> **Summary**
>
> In short, this example rigorously verifies the following phenomenon:
>
> 1. There are two local minima in sharp and flat regions, respectively. The distance between these minima is $p=O(1)$
> 2. A certain step-size depending on the problem can eliminate only a sharp minimum.
> 3. SGD and averaged SGD converge to a solution of smoothed function (which is a perturbation of flat minimum). And optimization accuracy can be arbitrarily small depending on the sharpness.
>
> Indeed, the distance between corresponding solutions of the original and smoothed objectives is O(\eta^2), but this example explains the SGD and especially averaged SGD are biased toward flat minimum by specifying all constants in our theory. (Moreover, we note that the solution of the smoothed objective is not within $O(\eta^2)$ of the sharp minimum of the original function.)
>
> We are confident our theory works and believe the main concerns have been addressed. Thus, we hope the reviewer has looked at our reply.

---

> ### Comment · Reviewer_EP51 · 2022-12-10
> **Questions on the Example**
>
> I appreciate the effort made by the authors to address my concern. Below are two questions I have after reading Appendix C.
>
> 1. In what sense Appendix C provides a separation? Is it theoretical or just experimental? A theoretical separation requires specifying a concrete problem satisfying the following two conditions: (a). the global minimizer of $f$ is different from that of its smoothed version, $F$; (b). the upper bound given by theorem 1 is smaller than the distance between the distance of the minimizer of $f$ and $F$.
> I cannot find (a) and (b) shown/proven anywhere in the current version of Appendix C.
>
> 2. Assuming $f$ has more than one local minimizer (otherwise it is not an interesting loss function for this study), Assumption 2 implies that $\eta$ must be larger than some constant depending on $f$, which means $\eta=O(1)$. Thus it is really not very meaningful to distinguish $O(\eta)$ and $O(\eta^2)$, as they are just all $O(1)$. Does this make sense?

---

> > ### Author Response · Authors · 2022-12-11
> > **Thank you for the response.**
> >
> > We address your concern below. Our example shows the separation theoretically. Even though there are two minima whose distance is about $p=O(1)$ but SGD and averaged SGD enter $O(\sqrt{\delta})$-and $O(\delta)$-neighborhoods of a flat minimum when $\delta$ is small (which represents flatness (sharpness) at one minimum and can be arbitrarily small). We also note that all assumptions are verified in Appendix C.3 with specific constants under the choice of step-size Eq. (10).
> >
> > 1. We would like to clarify the setup of the example.
> > The original objective has two local minima at $w=p$ and around $0 \sim w \in [-\delta,\delta]$. (Either of these could be a global minimum.) The distance between these two minima is at least $p-\delta$ but SGD and averaged SGD can enter $O(\sqrt{\delta})$-and $O(\delta)$-neighborhoods of $p$. (See the bounds $\sum \|v_t-v_*\| /T$ on p.20 for SGD and $\|E[\overline{v}\_T] -v_*\|$ on p.21.) This means the separation between sharp and flat minima; that is, it implies the bias of SGD and averaged SGD toward a flat minimum for the problem with the smaller $\delta$.
> >
> > 2. A required step-size $\eta$ for Assumption 2 depends on the problem setups (e.g., $\delta$). We note that $\delta$ represents a flatness (sharpness) at a minimum $w\sim 0$ in this example and $\eta = O(\delta)$ is sufficient to satisfy Assumption 2 as shown in Appendix C3 (see the paragraph **estimation of c**). That is, our example proves only sharp minimum can be eliminated with $\eta=O(\delta)$ when there is a separation of sharpness of minima. On the one hand, the sharpness at $w\sim 0$ depends on $O(1/\delta)$; and on the other hand, the sharpness at $w=p$ is $1$. Hence, the separation becomes larger as $\delta$ is small.
> > Moreover, averaged SGD can converge closer to a flat solution $w=p$ when $\delta$ is small (i.e.,  $\eta=O(\delta)$ vs $\sqrt{\eta} = O(\sqrt{\delta})$.)

---

> > > ### Author Response · Authors · 2022-12-11
> > > **Follow-up comment**
> > >
> > > In short, the example in the Appendix shows the separation between sharp and flat minima of the original function $f$ rather than the separation between a solution of $f$ and its continuous orbit of $F_\eta$. In other words, we consider a certainly large (but arbitrarily small, depending on the problem) step-size that eliminates an orbit of the sharp minimum, resulting in the bias towards a flat minimum. And a required step-size to eliminate a sharp minimum can be small if the sharpness is high (i.e., small $\delta$).

---

### Decision · Program_Chairs · 2023-01-20

**Decision:**

Reject

**Justification For Why Not Higher Score:**

Lack of support among reviewers.

**Justification For Why Not Lower Score:**

N/A

**Metareview: Summary, Strengths And Weaknesses:**

Summary: The manuscript studies the optimization and generalization trade-off caused by stochastic noise, step size, implicit bias, and sharpness of the minimum; and more importantly, why an averaged SGD with a large step size can improve the trade-offs and converge to a flat region more stably than SGD.

This paper was considered borderline by the reviewers (even after considering the rebuttal and updating their reviews). Reviewer vTYE recommended acceptance, but did not argue for it in discussion (and provided a fairly short review). All the other three reviewers gave the paper a 5. They acknowledged that it makes an interesting contribution to ICLR, providing a novel understanding of the implicit bias of averaged SGD. However, they felt it had some unclear points that needed to be addressed, in particular, as mentioned by reviewer EP51 and fQ6K, "whether there exists a suitable learning rate, such that the optimization-bias tradeoff gives non-trivial improvement compared to the empirical minimizer of the original training loss".